

# Among the world's smallest vertebrates: a new miniaturized flea-toad (Brachycephalidae) from the Atlantic rainforest

Luís Felipe Toledo[1], Lucas Machado Botelho[2],
Andres Santiago Carrasco-Medina[1], Jaimi A. Gray[3], Julia R. Ernetti[1],
Joana Moura Gama[4], Mariana Lucio Lyra[5], David C. Blackburn[3],
Ivan Nunes[6] and Edelcio Muscat[2]

[1] Laboratório de História Natural de Anfíbios Brasileiros (LaHNAB), Departamento de Biologia Animal, Instituto de Biologia, Unicamp, Campinas, São Paulo, Brazil
[2] Dacnis Project, Ubatuba, São Paulo, Brazil
[3] Department of Natural History, Florida Museum of Natural History, University of Florida, Gainesville, Florida, United States
[4] Departamento de Genética, Evolução, Microbiologia e Imunologia, Laboratório de Genômica Evolutiva, Departamento de Genética, Evolução, Microbiologia e Imunologia, Unicamp, Campinas, São Paulo, Brazil
[5] New York University Abu Dhabi, Abu Dhabi, Saadiyat Island, United Arab Emirates
[6] Laboratório de Herpetologia (LHERP), Instituto de Biociências, Campus do Litoral Paulista, Unesp, São Vicente, São Paulo, Brazil

Corresponding author
Luís Felipe Toledo,
toledosapo@gmail.com

## ABSTRACT

The genus *Brachycephalus* includes miniaturized toadlets with two distinct morphotypes: brightly colored species with a bufoniform phenotype and smaller, cryptic species with a leptodactyliform phenotype. The diversity of leptodactyliform species is still underappreciated, and we generally lack fundamental information about their biology. Recent sampling efforts, including DNA analyses and recordings of advertisement calls, have improved our understanding of this group. In the present study, we describe a new species of *Brachycephalus*, one of the smallest vertebrates known. This new species is distinguished from its congeners by a combination of morphological, bioacoustic, and genetic data. Despite being among the smallest frogs globally (the second smallest amphibian species), it exhibits skeletal traits typical of larger frogs, such as the presence of cranial bones that are lost or fused in other miniature frogs, including other *Brachycephalus*. Our description underscores how new discoveries within the megadiverse fauna of the Atlantic Forest—a rich biodiversity hotspot—can provide insights into phenotypic variation, including vertebrate body size. By describing this new species, we also aim to revisit the hypothesis that the type series of *B. hermogenesi* includes two species, potentially including individuals of the species described here.

## INTRODUCTION

The complex anatomical structures of vertebrates, including multiple and highly structured organs, puts limits on their body size. The largest known vertebrates include extinct archosaur reptiles, such as the colossal dinosaur *Argentinosaurus huinculensis* (likely >30 m in length: *Bonaparte & Coria, 1993*) and the prehistoric whale *Perucetus colossus* (weighing >300 tons: *Bianucci et al., 2023*). Among extant mammals, elephants and cetaceans are particularly notable for their size, with the Blue Whale (*Balaenoptera musculus*) being the heaviest and one of the largest animals ever (reaching >32 m in length and ~150 tons: *Sears & Perrin, 2009*). These gigantic creatures face physiological constraints, such as blood pumping, and morphological limitations, such as weight support, that impose a limit on their growth (*Vermeij, 2016*; *Goldbogen, 2018*).

Likewise, the smallest known vertebrates also present unique traits related to their extreme sizes, including the loss, reduction, and/or fusion of bones (*Trueb & Alberch, 1985*; *Hanken & Wake, 1993*; *Scherz et al., 2019*). These include freshwater cyprinid fish *Paedocypris progenetica* (the smallest mature individual reported was 7.9 mm in total length: *Kottelat et al., 2006*) and direct-developing frogs, such as *Paedophryne amauensis* (the smallest adult individual with a snout–vent length (SVL) of 7.0 mm: *Rittmeyer et al., 2012*) and *Brachycephalus pulex* (the smallest adult individual with a SVL of 6.5 mm: *Bolaños, Dias & Solé, 2024*). Among amphibians, the Neotropical genus *Brachycephalus* (pumpkin toadlets and flea toads; Anura: Brachycephalidae) is noteworthy for containing several of the smallest vertebrates, including species that reach maturity at less than 1 cm SVL (*Rittmeyer et al., 2012*; *Bolaños, Dias & Solé, 2024*). The skeletons of species of *Brachycephalus* exhibit bone fusions in both the skull and postcrania (*Rittmeyer et al., 2012*; *Izecksohn, 1971*; *Bornschein et al., 2016*), loss of skull bones (*Rittmeyer et al., 2012*; *Yeh, 2002*), fewer digits and phalanges (*Rittmeyer et al., 2012*; *Izecksohn, 1971*, *1988*; *Yeh, 2002*; *Pombal, Wistuba & Bornschein, 1998*), loss of vomerine or maxillary teeth (*Izecksohn, 1971*) that can sometimes be replaced by maxillary odontoids (*Ribeiro et al., 2017*), loss of middle ear bones and an external tympanum (*Goutte et al., 2017*), and having miniaturized semicircular canals (*Essner et al., 2022*). In addition, *Brachycephalus* species typically lack metacarpal, metatarsal, and subarticular tubercles (*Pombal & Izecksohn, 2011*), enervation connecting the inner ear to the brain (*Goutte et al., 2017*), a complete atrium septum, and a carotid body (*Carrasco-Medina et al., 2023*). All species are also thought to have direct-development with reduced fecundity and large eggs (*Trueb & Alberch, 1985*; *Hanken & Wake, 1993*). All of these traits are associated with the evolutionary reduction in body size; *i.e.*, miniaturization (*Hanken & Wake, 1993*; *Zimkus et al., 2012*).

The miniaturized body of these toadlets constrains their morpho-physiology, ecology, and life history, as well as impacting their detectability in the wild and taxonomic identification. These tiny toadlets are difficult to detect in the field because, apart from the brightly colored species ('bufoniform' species: *Lyra et al., 2021*), the smallest species are generally cryptic ('leptodactyliform' species: *Lyra et al., 2021*; or the '*B. didactylus* species group': *Ribeiro et al., 2015*) by matching the background coloration pattern (*Rebouças*

*et al., 2019*). Species with these major phenotypes are not monophyletic (*Condez, Haddad & Zamudio, 2020*; *Lyra et al., 2021*), and either the lack of or convergence of diagnostic traits resulting from miniaturization makes taxonomic identifications challenging (*Hedges, Duellman & Heinicke, 2008*; *Rittmeyer et al., 2012*). Despite these challenges, documenting the diversity of miniaturized vertebrates provides important opportunities for research on the anatomical evolution of vertebrates. For example, by understanding the lower limits of organ sizes, we gain insights into their operation and putative functions (*Hanken & Wake, 1993*; *Essner et al., 2022*).

Among the leptodactyliform species, a recent review made clear that *B. hermogenesi*, *B. sulfuratus*, and a still undescribed species (*Brachycephalus* sp.) are morphologically cryptic and can only be diagnosed through their advertisement calls (*Bornschein et al., 2021*). Furthermore, it was suggested that the type series of *B. hermogenesi* might contain individuals of a new species (*Bornschein et al., 2021*). Therefore, based on additional collections in the region where both species occur (*B. hermogenesi* and the new species), we describe a new flea-toad that is a leptodactyliform species of the genus *Brachycephalus* with adults that are among the smallest known vertebrates.

## MATERIALS AND METHODS

### Specimens and field sampling

We conducted fieldwork at the Projeto Dacnis private reserve (Ubatuba, São Paulo, 23°27′ 38.92″S, 45° 8′24.48″W, WGS84), located in the Brazilian Atlantic Forest Domain. The predominant vegetation cover in the region is formed mainly by secondary Alluvial Ombrophilous Dense Forest and Submontane Ombrophilous Forest. The annual mean temperature is 22.5 °C and the annual mean rainfall is 2,552 mm$^3$ (*Alvares et al., 2013*).

The specific site where the individuals were found, within the Projeto Dacnis reserve, was monitored from June 2021 to May 2022. Diurnal visits occurred twice each week and nocturnal visits once every 15 days. The area was sampled using visual and auditory search methods. The visual sampling consisted of careful removal and analysis of the leaf litter at the specific site and at different locations in the Projeto Dacnis reserve. In some cases, individual toadlets were observed calling, confirming them as adult males.

Collected specimens were deposited at the amphibian collection (ZUEC-AMP) of the Museu de Diversidade Biológica (MDBio), Universidade Estadual de Campinas (Unicamp), Campinas, state of São Paulo, Brazil, and Coleção de Anfíbios do Laboratório de Herpetologia (HCLP-A), Universidade Estadual Paulista, São Vicente, state of São Paulo, Brazil.

### External morphology and morphometry

We took 14 body measurements, following *Duellman (1970)*, *Heyer et al. (1990)*, and *Napoli (2005)*: snout–vent length (SVL), head length (HL), head width (HW), eye to nostril distance (END), interorbital distance (IOD), eye diameter (ED), nostril diameter (ND), upper arm length (UAL), forearm length (FAL), hand length (HAL), tibia length (also called shank or crus length; TBL), thigh length (THL). We also measured foot length, including the tarsus (FL) and without the tarsus (fL). We followed *Heyer et al. (1990)* for

morphological terminology of the snout shape. The measurements were taken with a digital caliper (precision 0.01 mm) and with a micrometric eyepiece attached to a stereo microscope (Leica S9). Examined specimens are listed in Appendix I.

### Internal anatomy: organs

One adult female (ZUEC-AMP 25273) was submitted to a dose of 2% lidocaine hydrochloride anesthetic solution, through administration *via* skin contact. Dissections were made immediately after the anesthesia procedure (*Iuliis & Pulerà, 2006*; *Sebben, 2007*). The individual was then immersed in an isotonic saline solution (NaCl 0.65%) to allow heart contractions and continuous blood flow as long as possible, facilitating the observation and description of blood vessels. The saline solution guarantees the maintenance of the morphophysiological characteristics of the organs and avoids unwanted reflexes that would otherwise prevent visualization. A Pasteur pipet was used to remove blood, skin, and/or candle wax that could interfere with the observations (*Sebben, 2007*). Photographs and videos were taken using a stereoscopic microscope (Leica S9), an Asus_A001D cell phone, and a Canon 5D Mark III camera coupled with a Sigma 105 mm lens. Visual files were deposited in MDBio with the voucher numbers ZUEC-PIC 699 (pictures) and ZUEC-VID 998 (videos).

### Internal anatomy: skeleton

We generated high-resolution X-ray computed tomography (microCT) scans at the University of Florida's Nanoscale Research Facility. We scanned the entire body of two specimens of the new species (ZUEC-AMP 24978 and 24979) and one of *Brachycephalus hermogenesi* (ZUEC-AMP 24980) using a GE v|tome|x m with a 240 kV microfocus x-ray tube with a tungsten target with the following settings: 60 kV, 110 mA (250 mA for 24,979), a 333 ms detector time (500 ms for 24,979) and no filters (see Table S2 for scanning parameters). All scans averaged three images per rotation with a skip of one, and a voxel resolution of 5–7 μm. We processed raw X-ray data using GE's datos|x software v.2.3 to produce tomogram images that were then viewed, segmented, and analyzed using VGStudioMax 2023.1 (Volume Graphics, Heidelberg, Germany). To produce figures showing the skeleton in VGStudio Max, we used the paintbrush tool with grayscale threshold, and the region growing tool, with enhanced lighting and shadows enabled to create a sense of depth. We use the organization of the osteological description following *Ribeiro et al. (2017)*, which uses the terminology of *Trueb, Diaz & Blackburn (2011)* and *Gómez & Turazzini (2016)*; we refer to the manual digits as I–IV rather than II–V to avoid confusion for most taxonomists. We compared the skeletons to existing datasets of species from the *Brachycephalus ephippium*, *B. pernix*, and *B. vertebralis* species groups as well as all flea-toads except for *B. pulex* and *B. puri*. Examined specimens for CT-scan data are in Appendix II.

### Bioacoustics

We used a Zoom H2n recorder with 16 bits of resolution and 44.1 kHz to record the calls. We conducted all sound analyses in Raven Pro v.1.6.3. We performed bioacoustic analyses

using the following spectrogram settings: window type = Hann, window size = 512 samples, hop size = 2.90 ms, overlap = 75%, DFT size = 1,024 samples, grid spacing = 43.1 Hz. We obtained the sound graphics using Raven software, and used the following spectrogram parameters: Hanning window, FFT = 512, and 70% overlap. We measured temporal and spectral parameters directly from the oscillogram and spectrogram, respectively, using parameters from Raven Pro, following *Köhler et al. (2017)*: call duration, inter-call interval, note duration, inter-note interval, note rate, number of pulses per note, dominant frequency, minimum frequency (frequency 5%), maximum frequency (frequency 95%), and frequency bandwidth (90%). We adopted the note-centered approach to describe the calls (*sensu Köhler et al., 2017*). All analyzed recordings of the new species are deposited at Fonoteca Neotropical Jacques Vielliard (FNJV), MDBio (FNJV 45418–30; 51341; 51346–8; not all males were collected).

To compare the advertisement call of the new species with that of *B. hermogenesi*, we made new recordings at its type locality, Parque Estadual da Serra do Mar, Núcleo Picinguaba, Ubatuba (FNJV 51343), as well as in the Projeto Dacnis reserve area (FNJV 51344–5), in Natividade da Serra (FNJV 51349), and reexamined the advertisement call from Estação Biológica de Boraceia, Salesópolis (FNJV 32769), all in the state of São Paulo, Brazil. Also, we used data provided by *Verdade et al. (2008)* for *B. hermogenesi* (FNJV 32769; from Salesópolis, SP), by *Condez et al. (2016)* for *B. sulfuratus* (FNJV 34498; from São Francisco do Sul, SC), and compared our data with that provided for *B. hermogenesi*, *B. sulfuratus* and the new species in *Bornschein et al. (2021)*. The concept of attenuated notes follows *Bornschein et al. (2021)*. Recordings were also used to confirm species distributions.

## Molecular data

We extracted total genomic DNA from ethanol-preserved muscle or liver tissues from individuals collected at Projeto Dacnis reserve, using both DNeasy Tissue Kits (Qiagen Inc., Valencia, CA, USA) or standard salt precipitation methods (*Lyra et al., 2021*). We amplified two overlapping fragments of the mitochondrially encoded 16S rRNA (16S). The polymerase chain reactions (PCR) settings and primers used are given in Table S3. All reactions were purified using Wizard® SV PCR Clean-Up System (Promega), following manufacturer's recommendation, or using a mix of thermosensitive alkaline phosphatase (FastAP) and Exonuclease I, as described in *Lyra, Haddad & de Azeredo-Espin (2017)*. The amplified products were sequenced in both directions by Macrogen Inc. (Seoul, South Korea), with a Big Dye v.3.0 Sequencing Kit (Applied Biosystems, Waltham, MA, US). We trimmed for quality, excluded primers sequences, and assembled consensus sequences using Geneious Prime® 2023.2.1 (Biomatters Ltd., Auckland, New Zealand). GenBank accession numbers and voucher information is available in Table S4.

For the phylogenetic inference, we aligned the newly generated sequences with representative 16S sequences of all species of *Brachycephalus* available in GenBank, including putative non-described species, and one individual of *Ischnocnema henselii* as an outgroup, totaling 99 individuals (Table S4). We aligned sequences with MAFFT v.7.427 (*Katoh & Standley, 2013*) using the E-INS-i algorithm in Geneious Prime® 2023.2.1. Phylogenetic analyses were performed under Bayesian inference (BI) using BEAST v2.7.1

(*Bouckaert et al., 2019*) under a birth-death tree prior, assuming a strict clock and using substitution rates of the GTR model. The analyses consisted of two independent runs, each with eight chains, for 100 million generations; parameters and trees were sampled every 1,000 generations, and we discarded the first 25% of iterations as burn-in. We assessed stationarity, convergence between runs, and effective sample sizes (>200) with Tracer (v.1.7; *Rambaut et al., 2018*). The log files of the independent runs were combined using LogCombiner (v.2.7; *Bouckaert et al., 2019*), and we extracted the majority-rule consensus tree in TreeAnnotator (V.2.7; *Bouckaert et al., 2019*). The resulting tree was visualized using FigTree 1.4.3 (*Rambaut et al., 2018*).

We estimated the uncorrected p-distances between the new species and the closest relatives using the 16S rRNA gene fragment limited by the primers 16Sar-L and 16Sbr-H ("Palumbi fragment"; *Palumbi et al., 1991*), using MEGA 11 (*Tamura, Stecher & Kumar, 2021*; *Stecher, Tamura & Kumar, 2020*) and treating missing data and gaps as pairwise deletions. Estimations were made using the alignment generated by MAFFT v.7.25. For comparative purposes, we also estimate uncorrected p-distances using the Palumbi fragment for all *Brachycephalus* species for which this fragment in available (*Palumbi et al., 1991*; Table S4).

## Species delimitation approach

We found two distinct *Brachycephalus* spp. vocalizations during field activities in Projeto Dacnis private reserve, municipality of Ubatuba, state of São Paulo: one was emitted by *B. hermogenesi* and the second belongs to a still undescribed species (E. Muscat and L. M. Botelho, 2021 personal observations). It also matches the bioacoustic findings reported by *Bornschein et al. (2021)* for the region. Our next step was to address different lines of evidence (vocalization, DNA sequence data, and morphology) to distinguish these two species. First, we recorded individuals of the genus *Brachycephalus* in the type locality of *B. hermogenesi* (Picinguaba, Ubatuba; *Giaretta & Sawaya, 1998*); no other congeners were found at that site. Since *B. hermogenesi* and the new species described here are morphologically similar but can be distinguished by their unique advertisement call and DNA sequences, we used this information to delimitate both species. To avoid misidentification, the comparisons were based solely on individuals that had been sequenced or had their advertisement call recorded.

The electronic version of this article in Portable Document Format (PDF) will represent a published work according to the International Commission on Zoological Nomenclature (ICZN), and hence the new names contained in the electronic version are effectively published under that Code from the electronic edition alone. This published work and the nomenclatural acts it contains have been registered in ZooBank, the online registration system for the ICZN. The ZooBank LSIDs (Life Science Identifiers) can be resolved, and the associated information viewed through any standard web browser by appending the LSID to the prefix http://zoobank.org/. The LSID for this publication is: urn:lsid:zoobank. org:pub:115413F3-9005-4006-863C-EA4BAD3C58AD, and for the new taxon is: urn:lsid: zoobank.org:act:18F93070-2CAE-4817-8CCD-48C0C5DA5210. The online version of this

work is archived and available from the following digital repositories: PeerJ, PubMed Central SCIE and CLOCKSS.

### Ethics and permits

Samplings and individuals' euthanasia were performed in accordance with relevant guidelines and regulations. Sampling permits were provided by Instituto Chico Mendes de Conservação da Biodiversidade (ICMBio, SISBio #51898-1). Specimen collection and deposition in scientific collections followed Brazilian animal care guidelines and were previously approved by the University of Campinas (Unicamp) animal care committee (CEUA IB/CLP #03/2020). The access to genetic information was also registered with the National System for the Management of Genetic Heritage and Associated Traditional Knowledge (SISGen #A74AD8B).

## RESULTS

*Brachycephalus dacnis* sp. nov. (Fig. 1).
*Brachycephalus* sp. (*Bornschein et al., 2021*).
LSID: urn:lsid:zoobank.org:act:18F93070-2CAE-4817-8CCD-48C0C5DA5210.

**Etymology–**The specific epithet name '*dacnis*' honors the Projeto Dacnis private reserve and NGO that has supported biodiversity research since 2010 in the municipalities of São José dos Campos, Miracatu and Ubatuba (where the new species was discovered), state of São Paulo, Brazil. The name is used as an invariable noun in apposition to the generic name.

**Holotype–**Adult male (ZUEC-AMP 24982) collected and recorded (FNJV 51341) at Projeto Dacnis reserve, municipality of Ubatuba, state of São Paulo, Brazil, on 18 May 2021 by A. Mariano (Fig. 1; Table 1). We also sequenced the DNA of this individual.

**Paratypes–**Eleven adults, only considered those confirmed by DNA and/or call traits: 03–09 August 2019 by R. Mitsuo and R.C. Menegucci (HCLP-A 267–68; heard, but not recorded); 10 March 2021 by E. Muscat and L.M. Botelho (ZUEC-AMP 24978; sequenced); 05 May 2021 by L.M. Botelho (ZUEC-AMP 24984; sequenced); 18 May 2021 by A. Mariano (ZUEC-AMP 24979; recorded); 24 May 2021 by E. Muscat and L.M. Botelho (ZUEC-AMP 24981; recorded and sequenced); 21, 22 February 2022 by L.M. Botelho (ZUEC-AMP 25270; 25272; 25274–75; sequenced); 14 April 2022 by E. Muscat, L. M. Botelho, L.A.O.S Ferreira (adult male; ZUEC-AMP 25612; Fig. 2; recorded). All paratypes were collected at the same locality as the holotype. Sex determination was not possible in most cases, except for the individual that was dissected (a female; see below) and the calling males.

**Diagnosis–**The new species is assigned to the genus *Brachycephalus* because of its miniature body size, fewer phalanges and toes than a typical frog, fingers and toe tips not expanded but apically pointed, and toes lacking circumferential grooves. The new species can be diagnosed from its congeners by the following combination of characters: (1) "leptodactyliform" body shape; (2) adult body length (SVL) smaller than 1 cm; (3) distinct

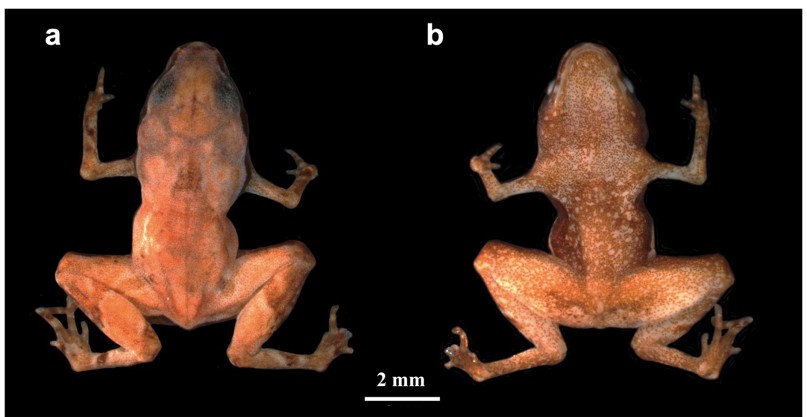

**Figure 1** *Brachycephalus dacnis* holotype (ZUEC-AMP 24982) adult male, SVL = 7.55 mm. (A) Dorsal and (B) ventral views.

**Table 1** Measurements of the holotype and adult paratypes (one adult that was dissected for internal anatomy was not measured) of *Brachycephalus dacnis*. Values presented as mean ± standard deviation (minimum–maximum) in millimeters.

| Trait | Holotype | Adult paratypes (*n* = 10) |
|---|---|---|
| SVL | 7.55 | 8.29 ± 1.03 (6.95–9.90) |
| HL | 1.78 | 2.59 ± 0.45 (1.87–3.18) |
| HW | 1.78 | 2.97 ± 0.55 (1.82–3.63) |
| END | 0.37 | 0.48 ± 0.23 (0.18–1.02) |
| IOD | 1.42 | 1.48 ± 0.32 (0.84–2.10) |
| ED | 0.49 | 0.94 ± 0.24 (0.50–1.30) |
| ND | 0.31 | 0.27 ± 0.08 (0.16–0.40) |
| UAL | 1.30 | 1.57 ± 0.22 (1.30–2.02) |
| FAL | 2.95 | 2.13 ± 0.30 (1.80–2.70) |
| HAL | 1.00 | 1.28 ± 0.23 (0.92–1.63) |
| TBL | 3.59 | 3.82 ± 0.48 (2.98–4.50) |
| THL | 4.00 | 3.77 ± 0.51 (3.25–5.00) |
| FL | 4.50 | 6.10 ± 0.41 (5.60–6.75) |
| fL | 2.75 | 3.16 ± 0.47 (2.35–4.00) |

and functional toes II and V; (4) presence of vestigial fingers I and IV; (5) distinct iris; (6) absence of dark markings on the skin over the pectoral region; (7) dark black or pale brown marbled venter with small white blotches in preserved specimens; (8) and advertisement call composed of one or two multi-pulsed (3–7 pulses) note with dominant frequency between 8.01 and 8.44 kHz, note duration between 0.03–0.08 s (when isolated), up to 0.41 s (when in pairs), and absence of attenuated notes.

**Holotype description**–Body slender and leptodactyliform (Fig. 1; SVL = 7.55 mm); head as wide as long (1.78 mm; Table 1), and one-fourth of total body length (HW/SVL = HL/SVL = 23.6%); snout short and slightly mucronate in dorsal view and between rounded and

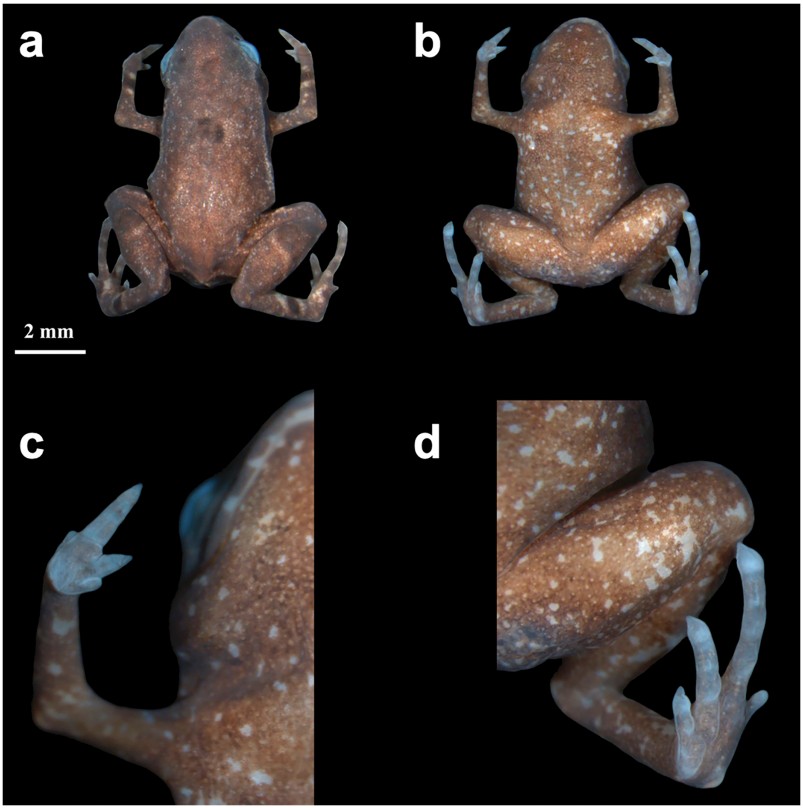

**Figure 2** *Brachycephalus dacnis* **paratype (ZUEC-AMP 25612) adult male, SVL = 7.89 mm.** (A) Dorsal view, (B) ventral view, (C) ventral view of hand, (D) ventral view of feet.

vertical in lateral view; nostrils oval, not protuberant and directed anterolaterally; *canthus rostralis* distinct; loreal region slightly concave; lips nearly sigmoid; eyes slightly protruding in dorsal and lateral view, and directed anterolaterally, 27.5% ED/HW and ED/HL (as HW = HL); tympanum indistinct; vocal sac not externally expanded; vocal slits present; tongue longer than wide, with posterior half free; choanae small and rounded; vomerine teeth absent. Forearm more than twice (2.3 times) length of upper arm length; hand length shorter than forearm or upper arm lengths; hands with all fingers distinct; fingers I and IV externally vestigial; fingers II and III robust; tips of fingers I and IV (when visible) rounded, fingers II and III pointed; relative finger length I ≤ IV < II < III; subarticular tubercles absent; inner and outer metacarpal tubercles absent. Leg relatively long, total leg length (TBL + THL + FL = 12.1 mm) 160% of SVL (7.6 mm); shank slightly shorter than thigh (SHL/SVL = 40.6% and THL/SVL = 47.5%); foot longer than tarsus and the shank; toe I externally absent; toes II, III, IV and V externally distinct and functional. Skin smooth without dermal ossifications. Measurements of holotype presented in Table 1.

**Color in life of the holotype**–The general coloration of the dorsal body surface is yellowish-brown. A dark brown stripe extends laterally from the tip of the snout to the flanks. Dorsal surfaces of arms and legs interrupted by dark brown stripes. An x-shaped or dorsal mark is present. The lateral view of the body is dark brown; the region overlying the

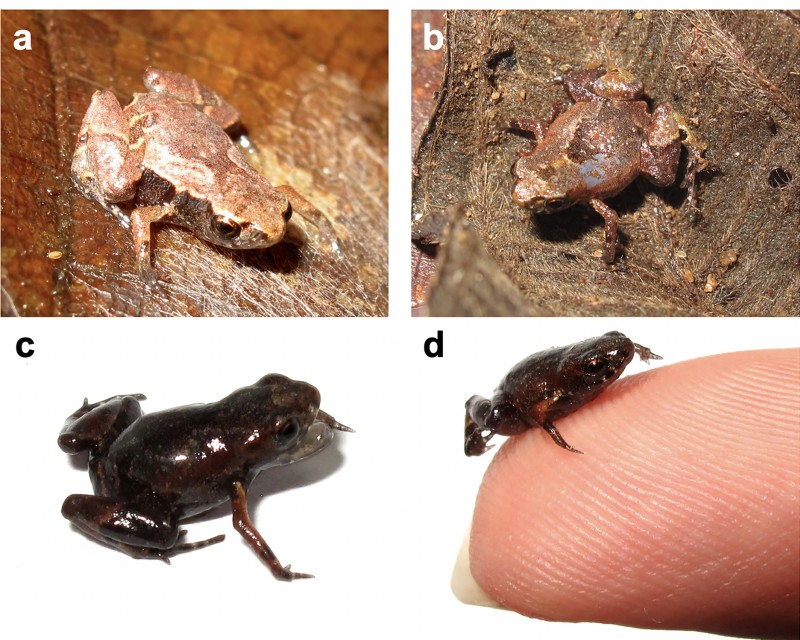

**Figure 3** *Brachycephalus dacnis* **paratype individuals found at the Projeto Dacnis private reserve, municipality of Ubatuba, state of São Paulo, Brazil.** (A) ZUEC-AMP 25272; (B) ZUEC-AMP 25274; (C and D) ZUEC-AMP 25275 displaying mouth-gaping behavior and the same individual on top of one herpetologist's fingertip.

upper jaw is dark brown with distinct white spots. The ventral body surface is brown with small white blotches. The ventral surface of the hands and feet are dark brown with white blotches over each phalanx of the fingers and toes. The pupil is black, and the iris is bronze.

**Color variation**–Color and ornamentation patterns in this species vary considerably among individuals. Some frogs have a leaf-like dorsal coloration, while others have a rock-like dorsal coloration (Fig. 3). We did not observe significant variation in the ventral pattern of the individuals.

**Color in preservative of the holotype** (less than 1 year in preservative)–The general coloration of the dorsal surface is reddish-brown (Fig. 1). The white spots overlying the upper jaw and ventral region can become pale cream; no *linea masculina* is visible. The stripes on the arms and legs become more evident. The extremities of the fingers and toes become pale cream.

**Internal anatomy: organs** (ZUEC-AMP 25273; female)–After cutting the skin, the transparent abdominal muscles were observed to have a slight dark-brown coloration (melanocytes). Beneath the musculature, the liver and small intestine were also observed with similar dark-brown pigmentation. However, it was more prominent in the intestine than in the ventral musculature and liver (Fig. 4A). After removing the abdominal muscles and opening the pericardium, we observed the heart to have melanocytes scattered throughout the atria, ventricle, *conus arteriosus*, and *truncus arteriosus*. After cutting the pectoral girdle in the medial region of the weakly mineralized procoracoid cartilages, we

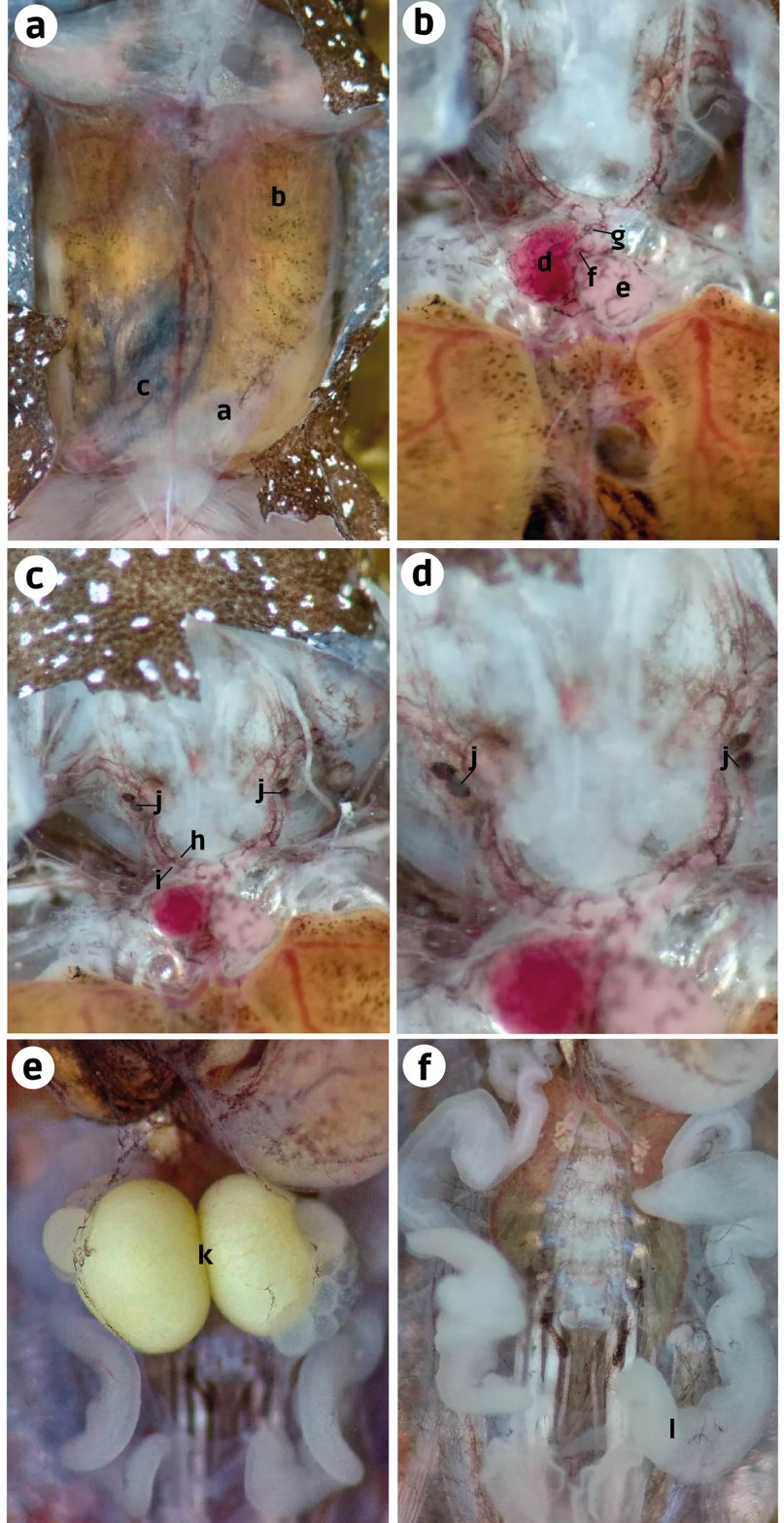

**Figure 4 Ventral dissection of *Brachycephalus dacnis* (ZUEC-AMP 25273) with the highlighted structures.** (A) Abdominal muscles, (B) Liver, (C) Small intestine; (D) Atrium, (E) Ventricle, (F) *Conus arteriosus*, (G) *Truncus arteriosus*, (H) Carotid artery, (I) Systemic artery, (J) Parathyroid gland, (K) Eggs, (L) Oviduct.

observed that the dark pigmentation extends from the cranial region of the heart to the carotid and systemic arteries. We did not see this pigmentation in the pulmocutaneous artery (Figs. 4B, 4C). A slight extension of this dark pigmentation occurs in the musculature lateral to the systemic and carotid arteries (Figs. 4B, 4C). Near the bifurcation of the internal and external jugular arteries and between the systemic and carotid arteries, there were two black, ovoid parathyroid glands on each side (Fig. 4C). Less black pigmentation was observed in the thoracic region; however, a few small spots were observed close to the white eggs and in the oviduct. The oviduct was transparent and between the left and right oviduct two mature eggs were observed, and the oviducts joined near the pubis (Figs. 4D, 4E).

**Internal anatomy: skeleton** (ZUEC-AMP 24978; undetermined sex). The skeleton of *B. dacnis* is broadly similar to that of other small species in the genus, including by lacking osteoderms and having unornamented dermal skull bones that are not co-ossified to the skin. This individual (Fig. 5) is osteologically mature with ossified mesopodials and complete, ossified distal long bones; the specimen is missing the right leg, which was removed for DNA analysis.

The skull is compact, slightly longer than wide, and lacking ornamented dermal roofing bones. The frontoparietals, sphenethmoids, prootics, exoccipitals, and parasphenoid are incompletely synostosed with each other and often have clear separations between the bones. There are well-defined and bony margins to both the optic fenestrae and prootic foramina. The premaxillae are broad, clearly separated from one another, and have teeth; each has a robust *pars dentalis*, and a robust alary process that is approximately as tall as wide and widely separated from the nasal. In ventral view, the maxillae are nearly straight and bear odontoids on the anterior third. The quadratojugals are thin and angled dorsally, and do not articulate with the maxillae. The pterygoids are slender, each with a long, straight, and forked anterior ramus that approaches but does not articulate with the adjacent maxilla, a short, subtriangular posterior ramus adjacent to but not articulating with the ventral ramus of the squamosal, and a short, broad medial ramus that is well separated from the prootic. Distinct neopalatines are not observable and likely not synostosed to the sphenethmoid. Distinct, small c-shaped vomers are present bordering the anterior margins of the choanae. Tall, thin, and curved septomaxillae are present at the anterior margin of the nasal capsule. The parasphenoid is triradiate with a long, rectangular anterior ramus. The squamosals are robust, and each has a prominent zygomatic ramus that is expanded ventrally as a sheet-like flange along its anterior margin and a long, slender posterior ramus that is adjacent but not fused to the prootic. The portions of the prootic are not completely synostosed and the broad fenestra ovalis contains a small, irregularly shaped operculum; a stapes (or columella) is not present. The posteromedial processes of the hyoid are ossified and slender. The arytenoid cartilages are not obviously mineralized.

There are eight distinct, procoelous, non-imbricating presacral vertebrae that are not fused to one another. The atlas lacks transverse processes and cotyles that are widely separated, a distance greater than the posterior diameter of the atlas centrum. Transverse

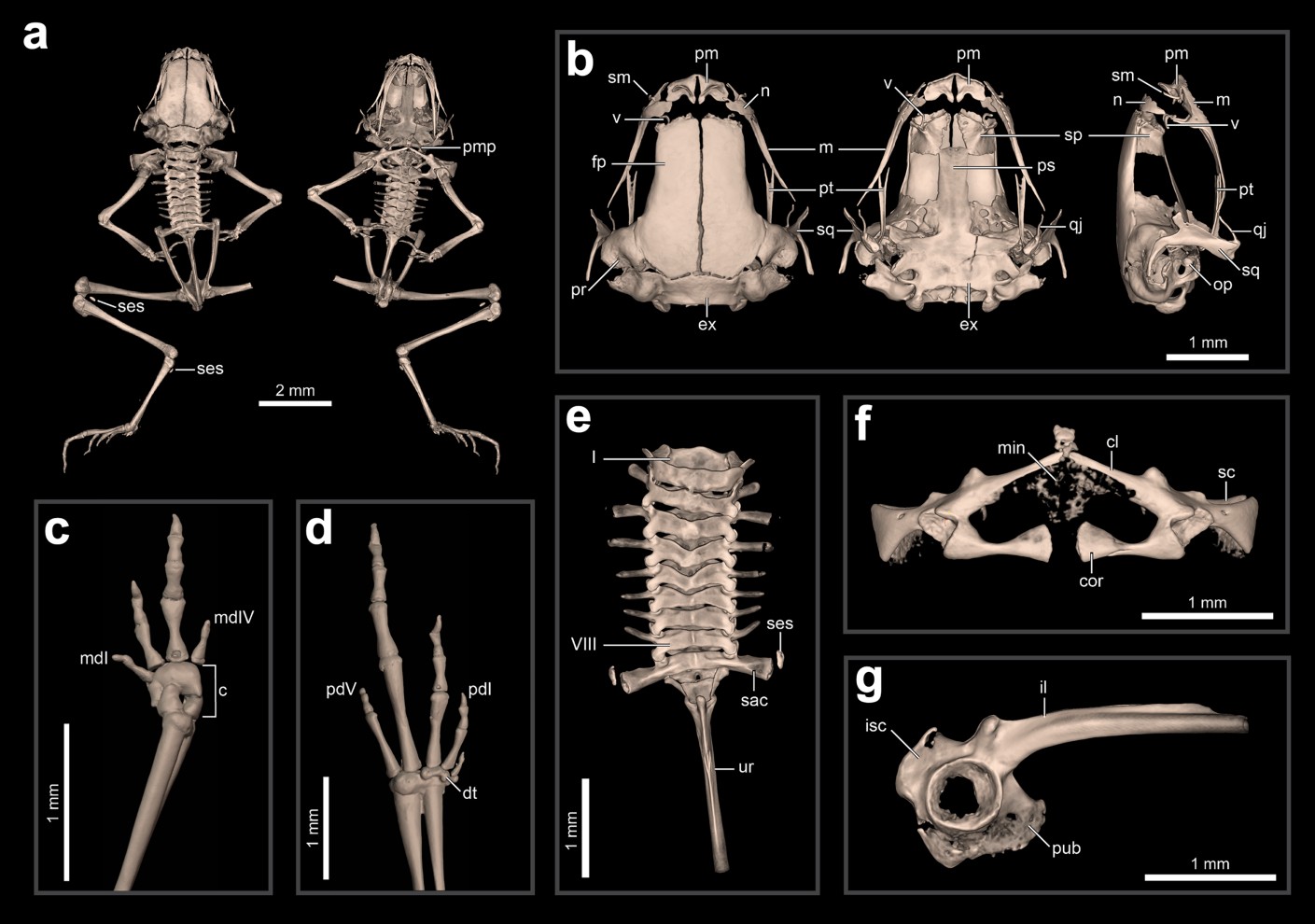

**Figure 5 Skeletal anatomy of *Brachycephalus dacnis* (ZUEC-AMP 24978; undetermined sex) based on CT-scans.** (A) The entire skeleton, with insets of (B) the skull (in dorsal, ventral, and right lateral views), (C) right hand and (D) left foot (both in dorsal view), (E) vertebral column (in dorsal view), (F) pectoral girdle (in ventral view), and (G) pelvic girdle (in right lateral view). Abbreviations: c, carpus; cl, clavicle; cor, coracoid; dt, distal tarsals; ex, exoccipital; fp, frontoparietal; il, ilium; isc, ischium; m, maxilla; mdI, manual digit I; mdV, manual digit V; min, mineralized cartilage; n, nasal; op, operculum; pd I, pedal digit I; pd V, pedal digit V; pm, premaxilla; pr, prootic; ps, parasphenoid; pt, pterygoid; pub, pubis; qj, quadratojugal; sac, sacrum; sc, scapula; ses, sesamoid; sm, septomaxilla; sp, sphenethmoid; sq, squamosal; ur, urostyle; v, vomer.

processes of presacral II are distinctly shorter than those of III–VIII. The sacrum is procoelous with stout transverse processes. A sesamoid at the distal transverse process of the sacrum is present at its articulation with the ilium. The urostyle bears a tall dorsal ridge that decreases in height posteriorly along the anterior two-thirds of the urostyle.

The pectoral girdle is not heavily ossified. The coracoids are expanded medially but do not meet at the midline. The coracoids are thin and articulate at the midline with a small, irregularly shaped element that may represent the omosternum. The scapula is short with a well-developed anterior process and a prominent supraglenoid foramen. The epicoracoid and procoracoid cartilages are weakly mineralized and do not form solid elements synostosed to the coracoid and clavicle.

The pelvic girdle is a robust element composed of synostosed ilium, pubis, and ischium; the publis and ischium appear to be incompletely ossified. The circular acetabulum is incompletely ossified but has well-defined margins. The shaft of the ilium is stout and mostly straight in both lateral and dorsal views, and has both a well-developed dorsal protuberance and a weakly developed dorsal crest. There is a broad ventral acetabular expansion comprising both the ilium and ossified pubis, but it is incompletely ossified.

The forearm is somewhat shorter than the humerus. The distal carpals (Element Y and II–V) are fused. The radiale and ulnare are large and subequal in size. The phalangeal formula for the manus is 1–2–3–1. There appears to be a single, minute ossified prepollex. There is one small palmar sesamoid. The tips of the terminal manual phalanges are arrow-shaped in digits II and III but blunt and globular in I and IV. The tibiofibula and femur are similar in length, and there is a small ossified fabella sesamoid near their articulation. There are two large distal tarsals and an ossified tarsal sesamoid near their articulation with the tibiofibula. The phalangeal formula for the pes is 1–2–3–4–2 and there is a single small ossified prehallux. Medially, there are two small plantar sesamoids, whereas laterally there are two large plantar sesamoids. The tips of the terminal pedal phalanges are arrow-shaped in digits I–IV but blunt in V.

**Calls and calling activity**–The new species is vocally active throughout the year, with peaks of activity when the humidity was higher than 90% (>50 males calling). The calling activity was mostly detected during daytime, with a higher number of individuals calling between 6:00 and 11:00 AM, from 3:00 to 6:00 PM, or anytime of the day after heavy rains. From August to April, we also heard calls during the night. Males frequently ceased calling for a few minutes after detecting the researcher's approach.

The advertisement call of *B. dacnis* (Fig. 6B) can be simple or complex, respectively containing one or two multipulsed (three to seven pulses) notes, emitted in mean intervals of 2.1 ± 0.32 s (1.67–2.69). Out of the 17 recordings made, only two presented calls with two notes. The mean note duration was 0.05 ± 0.01 s (0.03–0.08), and the call with two notes could reach up to 0.41 s. The call frequency varied from 6.89 (minimum measured frequency) to 8.96 (maximum measured frequency) kHz, while the mean dominant frequency was 8.22 ± 0.1 kHz (8.01–8.44) (Table 2).

**Phylogenetic inference and genetic distances**–Our phylogenetic analysis supported the distinction of *B. dacnis* from all the other members of the genus. All the specimens are clustered within a strongly supported clade (posterior probability, PP = 1.0). The new species is recovered as the sister species of *B. hermogenesi*, and this clade sister of *B. sulfuratus* + *B. pernix* species group (Figs. 6C; S1, Table S4). The phylogenetic inference recovered *B. pulex* sister to two diverse clades, one containing ((*B. puri* + two putative distinct species) + (*B. didactylus* + (*B. clarissae* + (*B. vertebralis* species group + *B. ephippium* species group)) and the other containing ((*B. hermogenesi* + *B. dacnis*) + (*B. sulfuratus* + *B. pernix* species group) (Figs. 6C, S1, Table S4). The 16S uncorrected genetic distance between the new species and *B. hermogenesi* ranged from 6.8% to 7.8% (mean 7.1%; Table S1). These distances are higher than distances found between species of *B.*

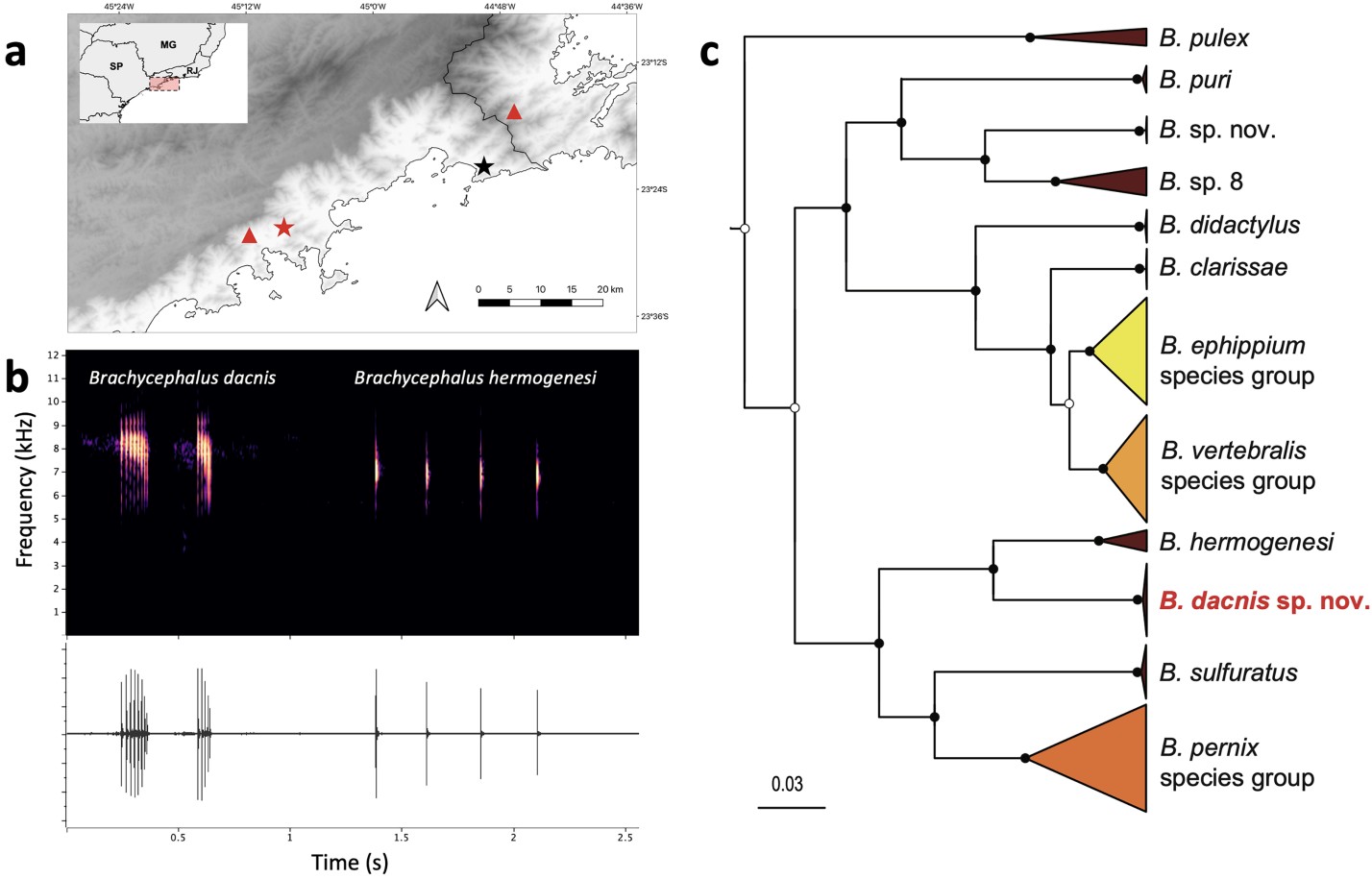

**Figure 6 Geographic distribution, advertisement calls and phylogeny.** (A) Type localities of *Brachycephalus dacnis* (red star) and *B. hermogenesi* (black star), and additional distribution records of *B. dacnis* (red triangles, based on call recordings published in *Bornschein et al., 2021*). This map was created using the free and open-source software QGIS, utilizing publicly available base data sources. (B) *Brachycephalus dacnis* holotype's advertisement call (FNJV 51341), recorded at Projeto Dacnis private reserve (type locality), Ubatuba, São Paulo (left) and *B. hermogenesi* topotype's advertisement call (FNJV 51343), recorded at Picinguaba (type locality), Ubatuba, São Paulo (right). (C) Schematic representation of the Bayesian phylogenetic inference of *Brachycephalus* based on the mitochondrially encoded 16S rDNA fragment. Black dots indicate fully supported clades (posterior probability, PP => 99). White dots indicate PP > 0.95. Node supports below pp = 0.95 are not shown. Complete tree in the supplement (Fig. S1).

*ephippium* and *B. vertebralis* species groups (3.2% to 5.9%, Table S1). The relationships between species within each of the species groups containing the bright colored and bufoniform species (*B. ephippium*, *B. vertebralis* and *B. pernix* species groups) are in general low supported (Fig. S1). Furthermore, some species were not recovered as monophyletic (*i.e.*, *B. pitanga* and *B. guarani*; *B. mariaeterezae* and *B. olivaceus*; Fig. S1) and the genetic distances estimated were extremely low, less than 1% (Table S1).

**Comparison with other species**–*Brachycephalus dacnis* is distinguished from all species of the *B. ephippium* (*B. atelopoide*, *B. darkside*, *B. ephippium*, *B. garbeanus*, *B. ibitinga*, *B. margaritatus*, and *B. rotenbergae*), *B. vertebralis* (*B. alipioi*, *B. bufonoides*, *B. crispus*, *B. guarani*, *B. herculeus*, *B. nodoterga*, *B. pitanga*, *B. toby*, and *B. vertebralis*), and *B. pernix* (*B. actaeus*, *B. albolineatus*, *B. auroguttatus*, *B. boticario*, *B. brunneus*, *B. coloratus*, *B. curupira*,

**Table 2 Advertisement call traits described for three flea-toads species from *Brachycephalus* genus.** Call traits present as mean ± standard deviation (range).

| | *Brachycephalus dacnis* six individuals (including the holotype) Dacnis, Ubatuba | *Brachycephalus dacnis* Corcovado, SP and Trilha do Corisco, RJ | *Brachycephalus hermogenesi* five individuals Ubatuba | *Brachycephalus hermogenesi* Several locations | *Brachycephalus sulfuratus* four topotypes São Francisco do Sul | *Brachycephalus sulfuratus* Several locations |
|---|---|---|---|---|---|---|
| Call duration (s) | 0.05 ± 0.01 (0.03–0.41) | - | 0.56 ± 0.14 (0.41–0.78) | - | 1.70 ± 0.10 (1.50–2.30) | - |
| Inter-call interval (s) | 2.1 ± 0.06 (1.67–2.69) | - | 2.70 ± 0.15 (2.52–2.91) | - | 4.5 ± 1.7 (3.1–7.4) | - |
| Number of notes per call | 1–2 | – | 2–6 | – | 4–6 | – |
| Presence of attenuated notes* | No | No | Yes | Yes | No | No |
| Note duration (s) | The same as call duration | - | 0.25 ± 0.04 (0.17–0.27) | - | 0.17 ± 0.01 (0.13–0.21) | - |
| Inter-note interval (s) | n/a | - | 2.70 ± 0.15 (2.52–2.91) | - | n/a | - |
| Note repetition rate (notes/s) | 19.58 ± 1.84 (12.55–29.41) | - | 4.12 ± 0.48 (3.58–4.88) | - | 0.2 ± 0.1 (0.2–0.3) | - |
| Number of pulses per note | 3–7 | 1–16 | 1–2 | 2–3 | 7–9 | 2–14 |
| Minimum frequency (Frequency 5%) (kHz) | 7.27 ± 0.12 (6.89–7.50) | - | 7.94 ± 0.08 (7.84–8.01) | - | 4.9 ± 0.3 (4.5–5.2) | - |
| Dominant frequency (kHz) | 8.22 ± 0.10 (8.01–8.44) | - | 8.38 ± 0.18 (8.18–8.61) | - | 6.6 ± 0.1 (6.5–6.7) | - |
| Maximum frequency (Frequency 95%) (kHz) | 8.62 ± 0.03 (8.44–8.96) | - | 8.91 ± 0.19 (8.70–9.13) | - | 9.1 ± 0.2 (9.0–10.7) | - |
| Frequency bandwidth (90%) (kHz) | 1.35 ± 0.12 (0.94–1.89) | - | 0.97 ± 0.18 (0.78–1.29) | - | n/a | - |
| References | Present study | *Bornschein et al. (2021)* | Present study | *Bornschein et al. (2021)* | *Condez et al. (2016)* | *Bornschein et al. (2021)* |

*B. ferruginus*, *B. fuscolineatus*, *B. izecksohni*, *B. leopardos*, *B. mariaeterezae*, *B. mirissimus*, *B. olivaceus*, *B. pernix*, *B. pombali*, *B. quiririensis*, *B. tabuleiro*, *B. tridactylus*, and *B. verrucosus*) species groups by the combination of its small adult body size, a leptodactyliform body form (*vs.* bufoniform), and a dark-brown coloration (*vs.* vibrant coloration such as greenish, yellowish, orange or reddish) (*Ribeiro et al., 2015*; *Lyra et al., 2021*). *Brachycephalus dacnis* is also distinguished from species of the *B. ephippium* and *B. vertebralis* species groups by its smooth skin, distinct frontoparietals, nasals, vomers, and parasphenoid that are not synostosed *via* a bony sphenethmoid (Figs. 5, 7), presence of a quadratojugal, robust anterior process of the pterygoid, lack of hyperossification in the skull, distinct coracoid and clavicle and lack of mineralized epicoracoid cartilages, lack of

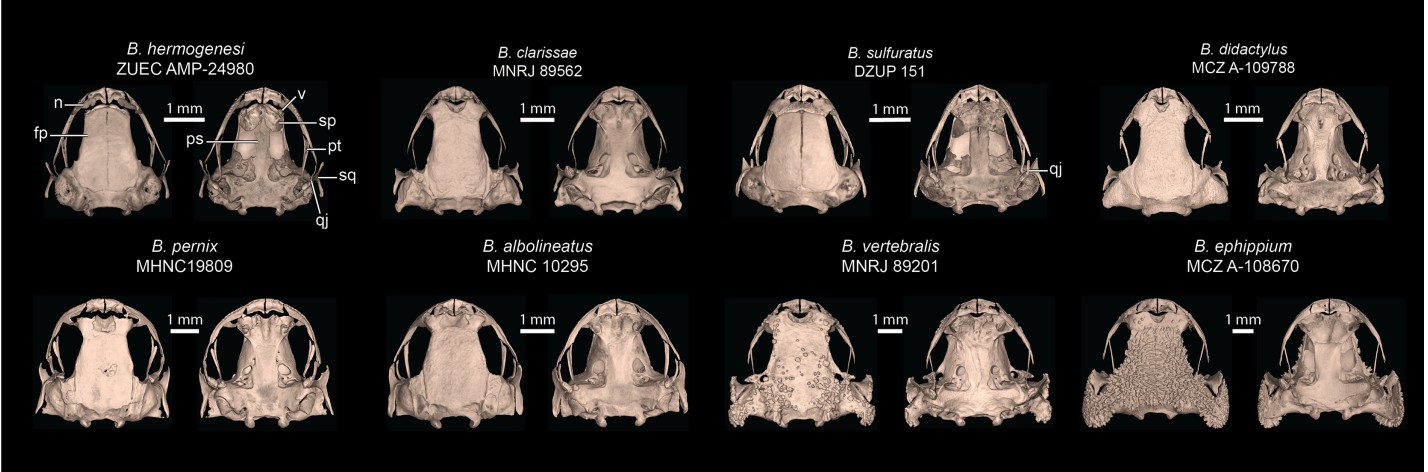

**Figure 7 Selected skulls of other *Brachycephalus* species.** Cranial elements labeled are discussed in comparisons of *B. dacnis* to other species. Top row is the leptodactyliform "flea toads", including *B. hermogenesi* (sister species of *B. dacnis*), *B. clarissae*, *B. sulfuratus*, and *B. didactylus*. Bottom row contains representatives of the three named species groups: *pernix* group (*B. albolineatus*, *B. pernix*), *vertebralis* group (*B. vertebralis*), and *ephippium* group (*B. ephippium*). Museum catalog numbers are listed below each species name. Abbreviations: fp, frontoparietal; n, nasal; ps, parasphenoid; pt, pterygoid; qj, quadratojugal; sp, sphenethmoid; sq, squamosal; v, vomer.

vertebral fusions, lack of bony plates in the axial skeleton, and presence of phalanges on all digits (*Lyra et al., 2021*; *Clemente-Carvalho et al., 2009*, *2012*; *Condez et al., 2014*, *2016*; *Haddad et al., 2010*; *Pombal, 2010*). From the other six leptodactyliform species, a.k.a. flea-toads (*B. didactylus*, *B. clarissae*, *B. hermogenesi*, *B. pulex*, *B. puri*, and *B. sulfuratus*), *B. dacnis* is distinguished by skin texture, toe morphology, coloration, and some skeletal traits. From *B. clarissae*, *B. dacnis* is distinguished by having a smooth dorsal texture (granular in *B. clarissae*), a distinct and functional toe V (vestigial in *B. clarissae*), lacking synostosis among cranial bones (sphenthmoids, parasphenoid, and frontoparietals synostosed in *B. clarissae*; Figs. 5, 7), phalanges on pedal digits I and V (lacking in *B. clarissae*), lacking mineralized epicoracoid cartilages (heavily mineralized and fused with pectoral bones in *B. clarissae*), and by having a distinct iris (indistinct in *B. clarissae*) (*Folly et al., 2022*). *Brachycephalus dacnis* has a distinct, and functional toe V (like *B. sulfuratus* and *B. hermogenesi*), whereas it is vestigial in *B. didactylus* (*Haddad et al., 2010*; *Giaretta & Sawaya, 1998*). Neither *B. dacnis* nor *B. hermogenesi* lack phalanges on the toes, whereas both *B. didactylus* (pedal digit count: 0, 2, 3, 4, 0) and *B. sulfuratus* (0, 2, 3, 4, 2) lack phalanges on one or more digit. In *B. didactylus*, there is a trapezoidal chest mark, which is absent in *B. dacnis* (*Almeida-Silva et al., 2021*), as well as a fully synostosed braincase (Fig. 7), a wire-like anterior process of the pterygoid, no quadratojugal, heavily mineralized epicoracoid cartilages fused with the pectoral bones, fused presacral vertebrae VI, VII, and VIII, and no phalanges on manual digit I or IV. From *B. sulfuratus*, *B. dacnis* is distinguished by the absence of an inverted V-shaped chest mark (*Condez et al., 2016*). Although this trait varies in *B. sulfuratus* (see *Bornschein et al., 2021*), it still can distinguish some individuals. From *B. sulfuratus*, *B. dacnis* is distinguished by its thin quadratojugal (Fig. 7), and medially expanded coracoid. From *B. puri*, *B. dacnis* is distinguished by having a distinct toe II (reduced in *B. puri*), by its dark black or pale brown marbled venter

with small white blotches (uniformly brown in *B. puri*), and by sometimes having an X-shaped dorsal mark (inverted V-shaped in *B. puri*) (*Almeida-Silva et al., 2021*). From *B. pulex*, *B. dacnis* is distinguished by the presence of vestigial fingers I and IV (absent in *B. pulex*), the presence of distinct and functional toe II and V (absent or vestigial, respectively, in *B. pulex*), and lacking a chest mark (presence of an inverted V-shaped chest mark in *B. pulex*) (*Almeida-Silva et al., 2021*). *Brachycephalus dacnis* and *B. hermogenesi* are morphologically similar, including in their skeletons (Figs. 5, 7), but can be recognized as distinct through acoustic and genetic traits.

The advertisement call of *B. dacnis* is distinguished from *B. hermogenesi* and *B. sulfuratus* by generally having fewer notes per call (one or two notes per call in *B. dacnis*, 2–6 in *B. hermogenesi*, 4–6 in *B. sulfuratus*), shorter note duration (≤0.08 s in *B. dacnis*; ≥0.17 s in *B. hermogenesi*, and ≥1.5 s in *B. sulfuratus*), and shorter call duration (≤0.08 s in *B. dacnis* in calls with one note and ≤0.41 s in calls with two notes; ≥0.41 s in *B. hermogenesi*, and ≥0.13 s in *B. sulfuratus*). *Brachycephalus dacnis* generally presents fewer pulses per note (3–7) than *B. sulfuratus* (7–11; *Condez et al., 2016*), and higher than *B. hermogenesi* (1–3) (see also *Bornschein et al., 2021*). The absence of attenuated notes (*sensu Bornschein et al., 2021*) in the calls of *B. dacnis*, distinguishes it from *B. hermogenesi* (which can present attenuated advertisement call notes) (see *Bornschein et al., 2021*). The dominant frequency of the advertisement call of *B. dacnis* overlaps with that of *B. hermogenesi*, but it is higher than that of *B. sulfuratus*. However, the minimum and maximum frequencies of the call of *B. dacnis* are lower and do not overlap with those of *B. hermogenesi*. Call parameters of these three species are presented in Table 2.

**Natural history**–We observed *Brachycephalus dacnis* occupying areas with dense vegetation, tall canopy, reduced sunlight incidence, a thick layer of litter, and moist soil. These areas lie between small hills, forming a valley. The leaf litter along the slopes lies in small clusters at the base of trees and creeping plants. These areas have a small stream nearby with a substrate composed of rock and sand. Water flow intensity varies with precipitation and can soak the adjacent soil. The new species also occurs in an area characterized by thinner vegetation, mainly shrubs, with more solar incidence, and a thick litter layer, and consequently, less moisture. In these sites, finding frogs was difficult because of their small size, cryptic coloration, cryptozoic habits, and the behavior of ceasing calling with any disturbance in their surroundings. After thorough searches, we found the individuals either between juçara palm (*Euterpes edulis*) roots or under rotten trunks. In these areas, *B. dacnis* was syntopic with *B. hermogenesi*. When approached, some individuals occasionally exhibited mouth-gaping (Fig. 3C). We also observed a 0.7 mm *Brachycephalus* cf. *dacnis* individual (not collected–but likely a *B. dacnis*, as no *B. hermogenesi* was found in the same area) jumping 21.8 cm (31 times its SVL) (ZUEC-VID 796).

We identified the stomach contents of two individuals. During dissection, we found one individual (ZUEC-AMP 25273) with four items in its stomach, identified to order: one Collembola, one Coleoptera, one Hymenoptera (Hexapoda), and one Acari

(Sarcoptiformes) (Fig. S2). In the microCT scan of another individual (ZUEC-AMP 24978), we identified one item, an Anobiinae beetle (Coleoptera: Ptinidae) (Fig. S2).

**Distribution**–*Brachycephalus dacnis* is known from its type locality, at Projeto Dacnis private reserve in Ubatuba, state of São Paulo (SP), Brazil, and two additional localities, based on acoustic records previously published in *Bornschein et al. (2021)*: Corcovado, municipality of Ubatuba, SP, and Trilha do Corisco, municipality of Paraty, Rio de Janeiro (Fig. 6).

## DISCUSSION

We describe a new species of *Brachycephalus*, that was first recognized by *Bornschein et al. (2021)* as a new entity, which ranks among the smallest known vertebrates (*Bolaños, Dias & Solé, 2024*). One of the individuals measured 6.95 mm, which is currently the second smallest adult vertebrate ever described, only larger than another individual of a different congeneric species (*Bolaños, Dias & Solé, 2024*). This individual was confirmed to be an adult, as it was one of the toadlets observed emitting advertisement calls. Although *B. dacnis* is at the lower bounds of anuran body size, it does not exhibit all the traits typically associated with extreme miniaturization in frogs. For example, among the flea-toads, *Brachycephalus pulex* has fewer functional toes, and *B. puri* and *B. didactylus* have more reduced or vestigial toes. The skeleton of *B. dacnis* is remarkable for not having the fusions and loss of elements typical of many miniaturized anurans (*Yeh, 2002*). The skeletons of *B. dacnis* and *B. hermogenesi* are nearly indistinguishable, but both differ substantially from skeletons of most other (even larger) species in the genus *Brachycephalus*. The genus *Brachycephalus*, especially *B. ephippium*, was previously considered a remarkable case of hyperossification associated with phylogenetic decreases in body size (*Trueb & Alberch, 1985*). The polyphyletic assemblage of flea-toads—the smaller *Brachycephalus* species— reveals that miniature species within the genus exhibit significant variation in the extent of skeletal fusions and losses. The phylogeny suggests that leptodactyliform species are among the earliest diverging species of the genus *Brachycephalus*. If so, then the "normalcy" of their skeletons suggests that many of the unique and remarkable traits of *Brachycephalus* evolved in association with the evolution of slightly larger body sizes, new body shape and conspicuous coloration within this genus. Thus, though extremely small, the skeleton of *B. dacnis* exhibits skeletal traits typical of larger frogs, such as the presence of cranial bones that are lost or fused in other miniature frogs, including other *Brachycephalus* (*Trueb & Alberch, 1985*). However, small body sizes in cryptically colored species might be under strong positive selection, as they could offer advantages for leaf litter frogs by reducing predator detectability and enhancing agility (*Blanckenhorn, 2000*).

This is the seventh flea-toad species described. It is likely that the diversity of flea-toads is underestimated due to their cryptic behavior (inhabiting the forest floor and having a cryptozoic lifestyle), small size (the smallest vertebrates on Earth), and overall color-matching of their habits (camouflage coloration). In addition, their lack of basic natural history data, such as recordings of advertisement calls, makes it difficult to identify new species (*e.g.*, *Bornschein et al., 2021*). Recent taxonomic studies have focused largely

on the brightly colored and larger *Brachycephalus* species (those in the species groups of *B. ephippium*, *B. pernix*, and *B. vertebralis*) (*e.g.*, *Nunes et al., 2021*; *Mângia et al., 2023*; *Folly et al., 2024*). A renewed focus on the tiny and cryptic species of *Brachycephalus* will likely reveal many more undescribed species at the lower bounds of body size in vertebrates.

Our phylogenetic analysis (Fig. S1) provides further evidence that *B. dacnis* represents a distinct evolutionary lineage and reveals significant genetic differences between flea-toad species (ranging from approximately 7% to 19%; Table S1), even among closely related species. However, it also underscores the remarkably low genetic divergence and weak support for certain species relationships, particularly within the *B. vertebralis* and *B. pernix* species groups (*e.g.*, *B. pitanga* and *B. guarani*; *B. olivaceus* and *B. mariaeterezae*; *B. coloratus* and *B. pernix*; see Fig. S1 and Table S1). These findings are consistent with previous studies (*Condez, Haddad & Zamudio, 2020*; *Folly et al., 2022*, *2024*; *Lyra et al., 2021*; *Mângia et al., 2023*; Fig. S6). *Pie et al. (2018*, *2019)*, using multiple nuclear markers and a multispecies coalescent model, provided a more detailed analysis of the phylogenetic relationships among some species in the *B. pernix* species group. Nevertheless, the posterior probabilities for some nodes were not always well supported in their results. Although their analysis supported the validity of these species (*Pie et al., 2019*), the methods employed could not distinguish between genetic structure due to species boundaries or structure due to population-level processes (*Sukumaran & Knowles, 2017*), as the authors noted in their discussion. Furthermore, they used only one individual per species, which prevented tests of species monophyly, and did not perform any test for gene flow.

Considering our results and previous studies, we suggest that the species status of some bufoniform species should be revised to clarify population diversity, species boundaries, and phylogenetic relationships. This revision will enhance our understanding of the processes that shape genetic and morphological diversity in the genus *Brachycephalus*. Notably, our findings highlight the urgent need for a taxonomic reevaluation of species within the *B. pernix* and *B. vertebralis* species groups, particularly those described based solely on a single diagnostic character, such as color variation or skin texture (*e.g.*, *Ribeiro et al., 2015*). These characteristics may represent population-level variation rather than distinct species (see also (*Monteiro et al., 2004*), for additional information on within-species color variation in *Brachycephalus* spp.). Future analyses should include a greater number of individuals per species and locality, the assessment of multiple nuclear genetic markers, population structure and phylogeographic analyses, the incorporation of bioacoustics and other phenotypic characters, and comprehensive sampling of all known species. This approach will provide a clearer framework for inferring phylogenetic relationships among all species within the genus, guiding future species descriptions as needed.

Additionally, taxonomic revisions should also encompass flea-toad species. For example, it is possible that the type series of *B. hermogenesi* is composed by individuals of *B. dacnis* (see *Bornschein et al., 2021*). Since the advertisement calls of those individuals were not recorded, and both species are morphologically cryptic, it would be advisable to investigate their DNA sequences. Recent molecular techniques for preserved specimens

(*i.e.*, historical DNA; hDNA) are available (*Straube et al., 2021*) and the description of *B. dacnis* will facilitate such resolution.

We observed individuals producing advertisement calls throughout the year. This may indicate a continuous reproduction strategy, as suggested by *Monteiro et al. (2018)* for *B. actaeus*. Calling activity may also be related to climate stability in Ubatuba, which provides a moist microhabitat in the leaf litter the whole year. We observed an increase in vocal activity in the wetter periods, with intensified activity after rains, including several males calling simultaneously throughout the day and some individuals calling at night, as observed by *Verdade et al. (2008)* and *Oliveira et al. (2011)* for *B. hermogenesi*. Most *Brachycephalus* species have diurnal activity, with calling emission strongly influenced by climatic conditions such as temperature, humidity, and rainfall (*Oliveira & Haddad, 2017*).

The substantial efforts for sampling and describing biodiversity reflect the continuously growing number of newly discovered species. By describing this new species in an integrative way, *e.g.*, by providing details of internal organs and the skeleton and descriptions of natural history (*e.g.*, diet, advertisement calls), we hope to facilitate the future description of more species in this surprisingly diverse and cryptic genus of tiny vertebrates.

## ACKNOWLEDGEMENTS

We thank Fernando Jacinavicius and Gabriel Biffi for the identification of the stomach contents; Elsie Laura Rotenberg for her efforts and struggle for the conservation of the Atlantic Forest; Alex Mariano dos Santos for his assistance in the field work; Dr. John Measey, Dr. Jennifer Vonk, Dr. Pete Binfield, and the anonymous reviewers for the significant time and effort they dedicated to reviewing this manuscript.

### Funding

Luís Felipe Toledo, Julia R. Ernetti and Mariana L. Lyra was supported by the São Paulo Research Foundation (FAPESP #2022/11096-8, #2020/02994-7) and the National Council for Scientific and Technological Development (CNPq #302834/2020-6). The funders had no role in study design, data collection and analysis, decision to publish, or preparation of the manuscript.

### Grant Disclosures

The following grant information was disclosed by the authors:
São Paulo Research Foundation: FAPESP #2022/11096-8, #2020/02994-7.
National Council for Scientific and Technological Development: CNPq #302834/2020-6.

### Competing Interests

The authors declare that they have no competing interests.

## Author Contributions

- Luís Felipe Toledo conceived and designed the experiments, analyzed the data, prepared figures and/or tables, authored or reviewed drafts of the article, and approved the final draft.
- Lucas Machado Botelho performed the experiments, authored or reviewed drafts of the article, and approved the final draft.
- Andres Santiago Carrasco-Medina performed the experiments, analyzed the data, authored or reviewed drafts of the article, and approved the final draft.
- Jaimi A. Gray performed the experiments, analyzed the data, authored or reviewed drafts of the article, and approved the final draft.
- Julia R. Ernetti performed the experiments, analyzed the data, authored or reviewed drafts of the article, and approved the final draft.
- Joana Moura Gama performed the experiments, analyzed the data, authored or reviewed drafts of the article, and approved the final draft.
- Mariana Lucio Lyra performed the experiments, analyzed the data, prepared figures and/or tables, authored or reviewed drafts of the article, and approved the final draft.
- David C. Blackburn performed the experiments, analyzed the data, prepared figures and/or tables, authored or reviewed drafts of the article, and approved the final draft.
- Ivan Nunes performed the experiments, authored or reviewed drafts of the article, and approved the final draft.
- Edelcio Muscat conceived and designed the experiments, authored or reviewed drafts of the article, and approved the final draft.

## Animal Ethics

The following information was supplied relating to ethical approvals (*i.e.*, approving body and any reference numbers):

Sampling permits were provided by Instituto Chico Mendes de Conservação da Biodiversidade (ICMBio, SISBio #51898-1). Specimens' collection and deposit in scientific collections followed Brazilian animal care guidelines and were previously approved by the University of Campinas (Unicamp) animal care committee (CEUA IB/CLP #03/2020). The access of genetic information was also registered at the National System for the Management of Genetic Heritage and Associated Traditional Knowledge (SISGen #A74AD8B).

## DNA Deposition

The following information was supplied regarding the deposition of DNA sequences:

The 100 GenBank sequences are available in the Supplemental Table. These sequences were described in a prior publication: MZ770736, MZ770737, MZ770738, MZ770739, MZ770740, MZ770741, MZ770742, MZ770743, MZ770751, MZ770752

## Data Availability

The raw data is available in the Supplemental Files.

### New Species Registration

The following information was supplied regarding the registration of a newly described species:

Publication LSID: urn:lsid:zoobank.org:pub:115413F3-9005-4006-863C-EA4BAD3C58AD.

Species LSID: urn:lsid:zoobank.org:act:18F93070-2CAE-4817-8CCD-48C0C5DA5210

### Supplemental Information

Supplemental information for this article can be found online at http://dx.doi.org/10.7717/peerj.18265#supplemental-information.

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
