# Peer review of "Among the world’s smallest vertebrates: a new miniaturized flea-toad (Brachycephalidae) from the Atlantic rainforest"

_PeerJ, doi:10.7717/peerj.18265_

## Round 0.1 · original submission · Major Revisions

As pointed out by reviewer 1, a relevant piece of the literature (Bornschein et al. 2021) is missing from your current manuscript. Incorporating this is likely to result in some re-analysis of your data as well as changes to your introduction and discussion. Hence, the decision given here is for Major Revision. I look forward to receiving a revised version of your manuscript taking into account the comments of both reviewers.

Bornschein, M.R., Ribeiro, L.F., Teixeira, L., Belmonte-Lopes, R., Moraes, L.A. de, Corrêa, L., Maurício, G.N., Nadaline, J. & Pie, M.R. 2021. A review of the diagnosis and geographical distribution of the recently described flea toad Brachycephalus sulfuratus in relation to B. hermogenesi (Anura: Brachycephalidae). PeerJ 9: e10983.

· Appeal

Appeal

The Section Editors believe that an incorrect decision was made


· · Academic Editor

Reject

The large number of criticisms by Reviewer 1 make this manuscript currently unpublishable, hence the Rejection. However I hope the feedback will help you make this manuscript potentially acceptable.

Reviewer 1 ·

Basic reporting

The paper introduces a new species of small anuran from the Brazilian Atlantic Forest, belonging to the genus Brachycephalus, specifically within the flea-toads group (referred to as the B. didactylus group by Ribeiro et al. [2015]). The authors provide a detailed description of the morphology of the new species and support their findings with phylogenetic analysis, aligning with contemporary expectations for species description articles.

<<Ribeiro, L.F., Bornschein, M.R., Belmonte-Lopes, R., Firkowski, C.R., Morato, S.A.A. & Pie, M.R. 2015. Seven new microendemic species of Brachycephalus (Anura: Brachycephalidae) from southern Brazil. PeerJ 3: e1011.>>

However, the authors omitted relevant literature, thus failing to capture all pertinent information regarding morphological and acoustic variation reported for both similar species already described and the species under their investigation. This oversight significantly undermines the conclusions of their work.

The species intended to be described by the authors has been previously mentioned in the literature by Bornschein et al. (2021), in an article published in PeerJ. While Toledo et al. cite this work in the synonymy of the new species (line 254; as Bornschein et al. 2020), they omit it from the references and fail to incorporate its data. Bornschein et al. (2021) provide two record points for the species in question, demonstrate sympatry with B. hermogenesi, describe part of its advertisement call, and offer a vocal diagnosis in comparison to B. hermogenesi and B. sulfuratus, the two flea-toads most similar to the new species.

<<Bornschein, M.R., Ribeiro, L.F., Teixeira, L., Belmonte-Lopes, R., Moraes, L.A. de, Corrêa, L., Maurício, G.N., Nadaline, J. & Pie, M.R. 2021. A review of the diagnosis and geographical distribution of the recently described flea toad Brachycephalus sulfuratus in relation to B. hermogenesi (Anura: Brachycephalidae). PeerJ 9: e10983.>>

The omission of literature results in several deficiencies in Toledo et al.’s work: 1) an inconsistent approach to the discovery history of the new species, 2) an outdated map of its geographic distribution (there are now three localities instead of just one), 3) an incomplete description of the advertisement call that fails to encompass all known individual variation (the sample size of recordings by Bornschein et al. [2021] exceeds that of Toledo et al.), 4) comparisons for the diagnosis of the new species that do not encompass all existing individual variation in similar species, and 5) failure to address the recommendation that the complex taxonomy of flea-toads requires comprehensive analyses.

“Combined with the fact that the B. didactylus group includes cryptic species, difficult or even impossible to identify in preservative, that occur or can occur locally in sympathy, we recommend a solid and broad review of the taxonomy of the group based on own analyses of large series of specimens and calls.” (Bornschein et al. 2021)

Additionally, the authors do not adequately characterize certain methods crucial for supporting their results, do not list the specimens/species they examined, do not personally assess subjective characteristics in compared specimens, and make errors in describing the advertisement call. This inhibits the accurate evaluation of the diagnostic characteristics proposed for the new species. Furthermore, no definitive difference between the new species and B. hermogenesi remains valid (please, see below under question #40, for example). I note that I observed the phylogeny and the robustness it added to the recognition of the population under study as a distinct entity. However, the entire taxonomy faces significant challenges in identifying diagnostic characteristics of well-supported clades, which, until then, must remain unnamed.

To rectify these issues, the authors should include literature, incorporate available data on the new species, conduct morphological analyses of other congeners, and conduct a robust review of characteristics within the B. didactylus group before proposing a new name in the literature. Below, I outline several observed points, though not exhaustive, that require attention to enhance the quality of the work.

Essential points
1) The raw data does not include the recordings (at least I did not find them), which could have been provided to enhance the quality of the review.

2) It is imperative that the manuscript includes a list of specimens identified based on recordings and/or genetics. After all, both the new species and B. hermogenesi are sympatric in the study area, and as I will demonstrate below, an objective diagnosis between them has not been provided. Thus, there is a serious risk that specimens collected in the study area may represent B. hermogenesi rather than the new Brachycephalus species. Moreover, Bornschein et al. (2021) noted that the type series of B. hermogenesi could potentially include specimens of the new Brachycephalus species. Therefore, this topic deserves careful attention.

<<Bornschein, M.R., Ribeiro, L.F., Teixeira, L., Belmonte-Lopes, R., Moraes, L.A. de, Corrêa, L., Maurício, G.N., Nadaline, J. & Pie, M.R. 2021. A review of the diagnosis and geographical distribution of the recently described flea toad Brachycephalus sulfuratus in relation to B. hermogenesi (Anura: Brachycephalidae). PeerJ 9: e10983.>>

3) Furthermore, it is equally imperative that the manuscript includes a list of examined recordings.

4) The introduction of the manuscript, as a whole, is oriented towards a study of miniaturization, but this is not the main scope of the work, nor the central focus of the discussions. The first paragraph particularly highlights how the theme has deviated from the topic at hand. I recommend focusing on the main topics and maintaining objectivity in the introduction and discussion sections.

5) The authors utilize the three phenetic groups proposed for the genus, but they seem to avoid doing so in a conventional manner by citing the literature. We are aware of a recent study demonstrating that the B. didactylus group is not monophyletic (Lyra et al. 2021). However, prior to this, the creation of three phenetic groups was proposed: B. ephippium, B. pernix, and B. didactylus (Ribeiro et al. 2015). In the manuscript, the authors use these three groups, explicitly citing the B. ephippium and B. pernix groups, but replacing the mention of the B. didactylus group with "flea-toads". I suggest that they adhere to the proposal from the literature or alternatively refrain from adopting any specific group, thus eliminating the mention of "flea-toads", including in the title of the work. This is because "flea-toads" is not a formally proposed group but rather a simple synonym for the B. didactylus group.

<<Lyra, M.L., Monteiro, J.P.C., Rancilhac, L., Irisarri, I., Künzel, S., Sanchez, E., Condez, T.H., Rojas-Padilla, O., Solé, M., Toledo, L.F., Haddad, C.F.B. & Vences, M. (2021). Initial phylotranscriptomic confirmation of homoplastic evolution of the conspicuous coloration and bufoniform morphology of Pumpkin-toadlets in the genus Brachycephalus. Toxins, 13(11), 816.

Ribeiro, L.F., Bornschein, M.R., Belmonte-Lopes, R., Firkowski, C.R., Morato, S.A.A. & Pie, M.R. 2015. Seven new microendemic species of Brachycephalus (Anura: Brachycephalidae) from southern Brazil. PeerJ 3: e1011.>>

6) Line 121-122: Examined specimens are listed in Appendix I.
Line 155-156: Examined specimens for CT-scan data are in Appendix I.
I couldn't find this appendix, and it's vital for the authors to include in the manuscript the list of examined material.

7) Line 159: We used a Zoom H2n recorder...
It's worth noting that the authors used a built-in microphone, which seems quite concerning. Yet another good reason for the recordings made to be uploaded as raw data to assess their quality. The issue of reduced recording quality affecting understanding and the ability to describe calls has already been mentioned (Bornschein et al. 2019). The authors should ensure that the recordings are of sufficient quality.

<<Bornschein, M.R., Rollo Jr., M.M., Pie, M.R., Confetti, A.E. & Ribeiro, L.F. 2019. Redescription of the advertisement call of Brachycephalus tridactylus (Anura: Brachycephalidae). Phyllomedusa 18(1): 3-12.>>

8) Why didn't the authors examine other available recordings, such as those from the Museu de História Natural Capão da Imbuia (MHNCI), Curitiba, Paraná?

9) Lines 178-180: All recordings were used to confirm species distributions, but for comparison purposes, we only used the recordings of the holotype of the new species and those made at the type localities of B. hermogenesi and B. sulfuratus.
The authors’ decision to restrict the sample size of recordings to such an extent is not justified in light of the need to measure intra-specific variation to better diagnose the new species. Additionally, if audio data serve to confirm the species’ distribution, why don’t they serve to characterize other characteristics besides distribution? I also emphasize that both B. sulfuratus and B. hermogenesi have a wide geographic distribution, and their distributions were revised with support from vocal and morphological analysis by Bornschein et al. (2021).

<<Bornschein, M.R., Ribeiro, L.F., Teixeira, L., Belmonte-Lopes, R., Moraes, L.A. de, Corrêa, L., Maurício, G.N., Nadaline, J. & Pie, M.R. 2021. A review of the diagnosis and geographical distribution of the recently described flea toad Brachycephalus sulfuratus in relation to B. hermogenesi (Anura: Brachycephalidae). PeerJ 9: e10983.>>

10) The authors did not specify which of the approaches by Köhler et al. (2017) they used to describe the advertisement call of the new species, whether a call-centered approach or a note-centered approach. In fact, there is a debate about which approach would be most appropriate, highlighting the importance of this detail.

<<Köhler, J., Jansen, M., Rodriguez, A., Kok, P.J.R., Toledo, L.F., Emmrich, M., Glaw, F., Haddad, C.F.B., Rödel, M-O. & Vences, M. 2017. The use of bioacoustics in anuran taxonomy: theory, terminology, methods and recommendations for best practice. Zootaxa, 908 4251(1), 1.124.>>

11) Lines 219-229: Species delimitation approach
This approach is unnecessary given the previous work that documented occurrence points, distribution, sympatry, and vocal diagnosis of the species that the present study aims to describe (Bornschein et al., 2021).

<<Bornschein, M.R., Ribeiro, L.F., Teixeira, L., Belmonte-Lopes, R., Moraes, L.A. de, Corrêa, L., Maurício, G.N., Nadaline, J. & Pie, M.R. 2021. A review of the diagnosis and geographical distribution of the recently described flea toad Brachycephalus sulfuratus in relation to B. hermogenesi (Anura: Brachycephalidae). PeerJ 9: e10983.>>

12) Line 257: Etymology
I’m not sure, but I believe that the allusion to the name of the type locality should be accompanied by the suffix 'iensis', perhaps “dacniensis”. The way the specific name is proposed, it only alludes to the meaning of dacnis, which is that of a bird from Egypt (https://www.wikiaves.com.br/wiki/sai-azul).

13) Line 266-267: Paratypes - Eleven adults, only considered those confirmed by DNA and/or call recordings (Table 1).
Is there an option for the authors to sex the individuals without conducting dissection, for example, through analysis of the presence / absence of vocal slit? Furthermore, the indication of “Table 1” suggests that the table will present the documentation method for the identity of the specimens collected, which is essential and has not been provided by the authors (see above, under question #2).

14) Line 266: Paratypes - Eleven adults
It is essential for the authors to indicate how they ensured that the specimens are adults. The indication that a specimen measuring 6.95 mm in SVL would be considered adult was used by the authors to present that the new species is one of the smallest vertebrates in the world (Table 1 and text). However, all Brachycephalus are small, and their measurements are even smaller as juveniles, obviously, and studies are cautious in discarding measurements of juveniles.

15) Line 280: Diagnosis - adult body length (SVL) smaller than 1 cm
I suggest exercising greater caution in diagnosing specimens as belonging to the new species if they measure less than 1 cm, considering that the largest measurement in the type series was so close, at 0.99 cm. It is expected that more collected specimens may provide material surpassing 1 cm.

16) Lines 280-281: Diagnosis - skin with diffuse mineralization but not organized into bony plates
The diffuse mineralization was not mentioned in any other part of the manuscript, neither in the description of the type series nor in the comparison between the species.

17) Line 281: Diagnosis - distinct and functional toes II and V
It would be desirable for the authors to eliminate the subjectivity from the description of the diagnostic parameter by using a measurement or proportion, for example.

18) Line 282: Diagnosis - presence of vestigial fingers I and IV
Another situation where it would be desirable for the authors to eliminate subjectivity from the description of the diagnostic parameter.

19) Line 282: Diagnosis - distinct íris
I apologize to the authors, but I do not understand what this definition of iris means.

20) Lines 282-283: Diagnosis - absence of dark markings on the skin over the pectoral region
It is desirable for a species to be defined by the characteristics it presents rather than the characteristics it does not present. In this case, it seems that the subsequent item in the diagnosis is the characteristic of ventral color.

21) Line 283: Diagnosis - dark black or pale brown marbled venter with small white blotches
In live specimens?

22) I recommend an assessment by the authors themselves of the coloration of specimens in preservative. It is expected that many species may not be diagnosable based on coloration in preservative, but it is an opportunity to present this reality to all researchers who will review scientific material in collections. Furthermore, researchers should also be clearly informed if the material that can be identified is limited to live specimens, which implies specific field procedures to document the coloration of live specimens and/or record specimens.

23) Lines 284-285: Diagnosis - and advertisement call composed of a single multi-pulsed (3.7 pulses) note with dominant frequency between 8.01 and 8.44 kHz, and note duration between 0.03.0.08 s.
In Figure 6a, the authors demonstrate that the advertisement call of the new species is composed of two notes.

24) Lines 304-305 Color in life of the holotype - The general coloration of the dorsal body surface is yellowish-brown (Fig. 1).
Figure 1 does not depict the coloration of the holotype in life, contrary to what the citation suggests.

25) Line 317 Color in preservative of the holotype (less than one year in preservative)
It is evident that the holotype was photographed shortly after preservation. It is desirable for the authors to replace the photo of the holotype with one taken currently and indicate how many months after fixation of the specimen this photo was taken.

26) Lines 402-403: The advertisement call of B. dacnis (Fig. 6b) is simple, containing a single and multipulsed (3 to 7 pulses) note
Please, see under questions #23 and 40, for details.

27) Lines 425-434 Comparison with other species -
I recommend that the authors provide the sample size considered for each comparative characteristic presented, in order to provide readers with the data support. This is particularly relevant in the case of bone characteristics, which vary greatly among individuals.

28) Lines 425-434 Comparison with other species - Brachycephalus dacnis is distinguished from all species of the B. ephippium (B. atelopoide, B. darkside, B. ephippium, B. garbeanus, B. ibitinga, B. margaritatus, and B. rotenbergae), B. vertebralis (B. alipioi, B. bufonoides, B. crispus, B. guarani, B. herculeus, B. nodoterga, B. pitanga, B. toby, and B. vertebralis), and B. pernix (B. actaeus, B. albolineatus, B. auroguttatus, B. boticario, B. brunneus, B. coloratus, B. curupira, B. ferruginus, B. fuscolineatus, B. izecksohni, B. leopardos, B. mariaterezae, B. mirissimus, B. olivaceus, B. pernix, B. pombali, B. quiririensis, B. tabuleiro, B. tridactylus, and B. verrucosus) species groups by the combination of its small adult body size, a leptodactyliform body form (vs. bufoniform), and a dark-brown coloration (vs. vibrant coloration such as greenish, yellowish, orange or reddish) (Lyra et al., 2021).
The authors are using a complementary proposal for group division, but the source has not been cited.

29) Lines 425-434: Comparison with other species - Brachycephalus dacnis is distinguished from all species of the B. ephippium (B. atelopoide, B. darkside, B. ephippium, B. garbeanus, B. ibitinga, B. margaritatus, and B. rotenbergae), B. vertebralis (B. alipioi, B. bufonoides, B. crispus, B. guarani, B. herculeus, B. nodoterga, B. pitanga, B. toby, and B. vertebralis), and B. pernix (B. actaeus, B. albolineatus, B. auroguttatus, B. boticario, B. brunneus, B. coloratus, B. curupira, B. ferruginus, B. fuscolineatus, B. izecksohni, B. leopardos, B. mariaterezae, B. mirissimus, B. olivaceus, B. pernix, B. pombali, B. quiririensis, B. tabuleiro, B. tridactylus, and B. verrucosus) species groups by the combination of its small adult body size, a leptodactyliform body form (vs. bufoniform), and a dark-brown coloration (vs. vibrant coloration such as greenish, yellowish, orange or reddish) (Lyra et al., 2021).
All the comparative data originate from Lyra et al. (2021). With certainty, no. The sources of the data need to be reported, unless the authors can provide their own analyses, which is desirable.

30) Lines 441-443 - Comparison with other species - From the other six leptodactyliform species, a.k.a. flea-toads (B. didactylus, B. clarissae, B. hermogenesi, B. pulex, B. puri, and B. sulfuratus),
As demonstrated above (question #5), the authors are using the B. didactylus group through a synonym (flea-toads), which is not necessary because this group was created based on phenetic characteristics and is being used by the authors precisely because of these agreeing phenetic characteristics.

31) Line 445 - Comparison with other species - a distinct and functional toe V (vestigial in B. clarissae)
The authors should remove subjectivity to make the diagnosis emphatic, direct, and not subject to interpretation (see above under question #17).

32) Line 449 - Comparison with other species - and by having a distinct iris (indistinct in B. clarissae)
I apologize once again for not understanding this definition of iris.

33) Lines 450-451 - Comparison with other species - Brachycephalus dacnis has a distinct, and functional toe V (like B. sulfuratus and B. hermogenesi), whereas it is vestigial in B. didactylus
The authors should remove subjectivity to make the diagnosis emphatic, direct, and not subject to interpretation (see above under question #17 and #30).

34) Lines 458-459- Comparison with other species - From B. sulfuratus, B. dacnis is distinguished by the absence of an inverted V-shaped chest mark (Frost, 2023)
Firstly, Frost (2023) is not a work that generated data to be cited in this case. This author merely compiled data. Secondly, Bornschein et al. (2021) demonstrated that this characteristic (inverted V-shaped) is not consistent in B. sulfuratus, rendering it unsuitable as a diagnostic feature.

35) Lines 458-460- Comparison with other species - From B. sulfuratus, B. dacnis is distinguished by ... a less robust quadratojugal (Fig. 7), and a coracoid that is more expanded medially.
Subjective terms like "less robust" and "more expanded" need to be avoided for the benefit of an objective diagnosis rather than an interpretative one.

36) Line 460 - Comparison with other species - From B. puri, B. dacnis is distinguished by having a distinct toe II
Once again, there is subjectivity in the diagnosis, which hinders the objective characterization of the new species.

37) Line 462 - Comparison with other species - and by sometimes having an X-shaped dorsal mark
“Sometimes” is something that cannot exist in a diagnosis: either the characteristic is supported by 100% of the specimens, or it does not serve for the diagnosis.

38) Lines 464-465 - Comparison with other species - vestigial fingers I and IV (absent in B. pulex), the presence of distinct and functional toe II and V (absent or vestigial, respectively, in B. pulex),
Same case of subjectivity in the diagnosis.

39) Lines 467-468 - Comparison with other species - hermogenesi are morphologically similar, including in their skeletons (Figs. 5, 7), but can be recognized as distinct through acoustic and genetic traits.
Genetics is not a character used in diagnosis. Here, it serves the purpose of reinforcing a difference, but this is not necessary.

40) Lines 469-473 - Comparison with other species - The advertisement call of B. dacnis is distinguished from B. hermogenesi and B. sulfuratus by having few notes per call (1 note per call in B. dacnis, 2.4 in B. hermogenesi, 4 or more in B. sulfuratus), shorter note duration 0.08 s in B. dacnis; 0.17 s in B. hermogenesi, and 1.5 s in B. sulfuratus), and shorter call duration 0.08 s in B. dacnis; 0.41 s in B. hermogenesi, and 0.13 s in B. sulfuratus).
Here, the authors attempted to present diagnostic differences but failed to achieve the goal due to the lack of specification regarding the description approach of the call (see above under question #10) and misinterpretation regarding the number of notes in the call of the new species (see above under question #23). The approach of the description is so relevant for the interpretation of results that a comparative table of call characteristics of Brachycephalus sp. nov. (= B. dacnis), B. sulfuratus, and B. hermogenesi was produced by Bornschein et al. (2021).

I present a "table" below with the number of notes and pulses for Brachycephalus sp. nov. (= B. dacnis), B. sulfuratus, and B. hermogenesi based on the data from Bornschein et al. (2021), according to the call-centered approach:

Parameter B. dacnis B. sulfuratus B. hermogenesi
Number of notes per call 1 1 1
Number of pulses per note 1-16 1-14 1-3

These data show overlap, and it is implied that notes with fewer pulses will be shorter and those with more pulses will be longer, suggesting overlap in the other two parameters indicated by the authors: call duration and note duration (representing the same parameter according to the call-centered approach). Thus, there is no diagnosis in the manuscript that supports differentiation between the proposed new species and B. hermogenesi.

41) Lines 473-474 - Comparison with other species - The multipulsed nature of the note of B. Dacnis distinguishes it from that of B. hermogenesi, which has indistinct pulses.
That's not true. Just examine Bornschein et al. (2021) to see how B. hermogenesi presents perfectly distinguishable pulses.

<<Bornschein, M.R., Ribeiro, L.F., Teixeira, L., Belmonte-Lopes, R., Moraes, L.A. de, Corrêa, L., Maurício, G.N., Nadaline, J. & Pie, M.R. 2021. A review of the diagnosis and geographical distribution of the recently described flea toad Brachycephalus sulfuratus in relation to B. hermogenesi (Anura: Brachycephalidae). PeerJ 9: e10983.>>

42) Lines 474-479- Comparison with other species - Brachycephalus Dacnis generally presents fewer pulses per note (3.7) than B. sulfuratus (7.11; Condez et al., 2016). The dominant frequency of the advertisement call of B. dacnis overlaps with that of B. hermogenesi, but it is higher than that of B. sulfuratus. However, the minimum and maximum frequencies of the call of B. dacnis are lower and do not overlap with those of B. hermogenesi. Call parameters of these three species are presented in Table 2.
This section is comparing characteristics that are not diagnostic. Therefore, they confuse the reader. The authors should present comparisons of diagnostic characteristics instead of shared ones.

43) I suggest producing a specific figure to support the description of the species’ call.

44) Line 501-502: Brachycephalus dacnis is known from its type locality, at Projeto Dacnis private reserve in Ubatuba, state of São Paulo, Brazil (Fig. 6).
As mentioned above, Bornschein et al. (2021) presented other records for the new species.

Secondary points
45) I suggest a title that includes the name of the genus to which the new species belongs.

46) The abstract does not reflect the reality of the work. It should address the results and discussions. I suggest including the diagnostic characteristics.

47) Line 31-32: The diversity of the leptodactyliform species is underestimated and we generally lack basic information about their biology.
The diversity of this group is as underestimated as, or even less so than, that of the B. ephippium group. The claim that cryptic coloration could be a reason doesn't hold, as even the colorful Brachycephalus species are difficult to spot and collect.

48) Line 83-84: and are not abundant in the forest litter (Rebouças et al., 2019).
Brachycephalus populations can reach densities of less than 1 individual occurring per 1 m2, resulting in population estimates of millions of individuals (e.g., Bornschein et al. 2019a,b)

<<Bornschein, M.R., Pie, M.R. & Teixeira, L. 2019a. Conservation status of Brachycephalus toadlets (Anura: Brachycephalidae) from the Brazilian Atlantic Rainforest. Diversity 11(9): 150.

Bornschein, M.R., Teixeira, L. & Ribeiro, L.F. 2019b. New record of Brachycephalus fuscolineatus Pie, Bornschein, Firkowski, Belmonte-Lopes & Ribeiro, 2015 (Anura, Brachycephalidae) from Santa Catarina state, Brazil. Check List 15(3): 379–385.>>

49) Lines 98-99: The predominant vegetation cover in the region is formed mainly by secondary Alluvial Ombrophilous Dense Forest and Submontane Ombrophilous Forest (Veloso et al., 1992).
Veloso et al. (1992) did not state that the study site had the indicated vegetation. They provided information to support the characterization of the vegetation, which was done by the authors.

50) Lines 102-107 of Specimens and field sampling
The authors present field efforts spanning several hours, yet there are no results or discussions derived from this effort. This appears to be a method for species inventory, which doesn’t seem relevant to the current study.

51) Lines 108-111: Collected specimens were deposited at the amphibian collection (ZUEC-AMP) of... and Coleção de Anfíbios do Laboratório de Herpetologia (HCLP-A), Universidade Estadual Paulista, São Vicente, state of São Paulo, both in Brazil.
There is no indication of the species deposited in the HCLP-A in the manuscript.

52) Line 164: FFT = 512
Why not FFT 256, which allows for better perception of the pulses?

53) Lines 396-397: Calls and calling activity - The new species is vocally active throughout the year, with peaks of activity when the humidity was higher than 90 % (> 50 males calling).
Would it be possible for the authors to provide temperature data?

54) Lines 399-400: From August to April, we also heard calls during the night.
Would it be possible for the authors to provide details (hour, temperature, humidity)?

55) Lines 430 Comparison with other species - B. mariaterezae
I suggest that the authors indicate the source for the spelling.

56) Lines 492: We also observed a 493 0.7 mm B. dacnis individual (not collected) jumping 21.8 cm
Without genetic or vocal analysis, it is not possible to determine the identity of this observed individual. The observation should be reported as belonging to a Brachycephalus sp.

Experimental design

Please see above.

Validity of the findings

Please see above.

Reviewer 2 ·

Basic reporting

In general, the text is well-referenced. There is just one aspect in the introduction that slightly concerns me. I understand that there's a trend in current scientific productions to address issues from a broader perspective, in order to allow for the widest possible reach of our publications. I also understand that, conversely, this has been reducing the relative scope of areas considered "classic" or "descriptive," such as taxonomy. However, there are elements that, in my opinion, need to be present in the context of the description of a species. In describing a Brachycephalus species, the authors frame the introduction around vertebrate miniaturization, which undoubtedly adds appeal and breadth to the text. However, the introduction to the genus is almost exclusively concerned about the size and the reduction of structures. I missed a broader contextualization of the group per se. For example, it's mentioned that there are two phenotypes in the group: the leptodactyliform cryptic flea toads and the bufoniform bright colored species. It would be useful to add that neither of the phenotypes is considered monophyletic (e.g., https://doi.org/10.1093/biolinnean/blz200), and perhaps to direct the text to treat in more detail the phenotype to which the new species belongs. Additionally, given that several other Brachycephalus species occur in the region (e.g., see the map in https://doi.org/10.1371/journal.pone.0244812), it would be interesting to provide some commentaries about the well-documented microendemism of the genus (e.g., https://doi.org/10.7717/peerj.2490). In fact, type localities of B. dacnis sp. nov. and B. hermogenesi (recovered as sister of the new species) are less than 50km apart (accordingly Fig. 6).

Experimental design

The manuscript provides a carefully prepared and well-elaborated species description that embraces a taxonomically integrative effort and effectively bridges the knowledge gap. However, as I previously mentioned, there is a lack of taxonomic information in the Introduction guiding the reader towards the taxonomy/systematics of the genus Brachycephalus or the flea toad group. I therefore feel that the knowledge gap is not clearly stated throughout the introduction. Nonetheless, this remains my sole observation.

Validity of the findings

The authors have effectively presented the data, making it available through appropriate means such as museum collections, Morphosource, Sketchfab, and Genbank. I would like to see further discussion about the advertisement call. It's noteworthy that has been documented intraspecific variation in the morphological traits used to differentiate between B. sulfuratus and B. hermogenesi (https://doi.org/10.7717/peerj.10983), being the advertisement call a significant tool in distinguishing between these species. Considering that B. dacnis sp. nov. and B. hermogenesi are in a similar situation, I believe it would be beneficial to reference Bornschein et al. (2020) and offer some commentary on this topic. As far as we are aware, the structure of the advertisement call appears to play a pivotal role in flea toad taxonomy. The conclusions are well articulated and appropriate given the presented results.

Additional comments

254 Brachycephalus sp. (Bornschein et al., 2020): This paper is not listed in the references.
347 incompletely synostosed with each other and have often have clear separations between the: Please remove the first “have”
458-459 From B. sulfuratus, B. dacnis is distinguished by the absence of an inverted V-shaped chest mark (Frost, 2023): The reference doesn't make sense. I suggest changing it to http://doi.org/10.11646/zootaxa.4083.1.2
531-533 Recent taxonomic studies have focused largely on the brightly colored and larger Brachycephalus species (those in the species groups of B. ephippium, B. pernix, and B. vertebralis) (Frost, 2023). The reference doesn't make sense. I suggest changing it to the original taxonomic studies you are referring to.
540-541 mariaterezae: mariaeterezae

---

## Round 0.2 · Minor Revisions

I must first start by apologising for the long delay and change of editor for this manuscript. The previous editor solicited the reviews that you will see below, and all other reviewers that were approached refused to review your ms.

Due to this unusual circumstance, and the conflicting reviewers' recommendations, I have read and also reviewed your ms. Although my decision is Minor Revisions, there is more required than the very small changes this may suggest. My reading of the manuscript revealed that your co-authors had not read this manuscript prior to submission as you asserted. The ms (and in particular the abstract) has many grammatical faults that your co-authors could not have read and left unchecked. Please ensure that they read your next revision AND make all necessary grammatical corrections. Failure to do so will reflect badly on them and result in another round of revisions that I'd rather avoid.

Reviewer 1 has requested a lot of changes, but I don't agree with all of these. However, I would like you to address the following points:

- The reviewer calls into question your assertion that your specimen was adult. To address this comment, I would like you to add a paragraph to the discussion including the reasons for your assertion and the implications for any uncertainty in your interpretation. You do not need to follow the comments of the reviewer, however I'd like an honest appraisal of this issue.
- The reviewer calls for a reanalysis of your acoustic data. I do not think this is necessary.

The remaining reviewer comments are minor and you should consider them. I think it would also be good to acknowledge all reviewers, as they have put a lot of time into your ms.

Reviewer 1 ·

Basic reporting

Dear Dr. Hedrick

I noted that the manuscript has been revised in numerous aspects, and I would like to congratulate the authors for their efforts.
I would like to begin my review by addressing three general aspects. 1) Firstly, I observed that some of the authors’ responses in their reply document deviated from the expected cordiality in peer interactions. For instance, the statement that “the reviewer is trying to be sneaky” reflects a tone less professional and more accusatory. The authors’ use of the specific name “mariaeterezae,” which deviates from the spelling of the original description, was proposed as a new spelling by Segalla et al. (2021) in a work that includes L. F. Toledo, the first author of the current manuscript. This is why I requested that they cite the source to avoid misinterpretation of the spelling as erroneous. 2) Secondly, in their responses to my questions #19 and #32, the authors’ comments imply that I do not understand the concept of an iris. I would like to reiterate my earlier suggestion that the authors provide a more detailed description of the iris characteristics of the new species. Simply stating that the iris is distinct does not sufficiently describe the color pattern it exhibits. For example, if a Brachycephalus with reddish irises were collected, the description of the iris for B. dacnis would not clarify whether this iris is similar or different. Providing this detailed information would be beneficial for future readers in identifying new specimens. It is also important to highlight that the golden-spotted iris of B. curupira is the sole morphological trait that differentiates this species from B. brunneus, which has an indistinct iris. 3) Lastly, it is customary for authors to acknowledge the reviewers for their contributions during the review process.

Reference
Segalla, M; Berneck, B.; Canedo, C.; Caramaschi, U.; Cruz, C.A.G.; Garcia, P. C. A.; Grant, T.; Haddad, C. F. B.; Lourenço, A. C.; Mangia, S.; Mott, T.; Nascimento, L. Toledo, L. F.; Werneck, F.; Langone, J. A. (2021) List of Brazilian amphibians. Herpetologia Brasileira, v. 1. https://zenodo.org/records/4716176

There are, in my opinion, two major problems that still need to be addressed in the manuscript: 4) the determination of the 6.95 mm specimen as an adult, and 5) the bioacoustics analysis. I will develop each of these issues in detail and then offer comments on other aspects of the manuscript.

4) The 6.95 mm specimen identified as an adult.
The classification of the 6.95 mm specimen as an adult forms the basis for the authors’ claim that B. dacnis is the second smallest vertebrate in the world. In my previous review (#13 and #14), I raised concerns about the sex determination and age assessment of this specimen, and the authors responded by stating that the specimen vocalized and chose not to confirm its sex by examining the presence of a vocal slit due to practical difficulties. While I acknowledge the challenges involved and agree that vocalization suggests the specimen is an adult, there is a critical issue that needs to be addressed. Condez et al. (2017) demonstrated that not all specimens identified as males by Ribeiro et al. (2015) were actually males, revealing that we cannot assume that the specimen heard vocalizing is the same one that we collect. Brachycephalus can occur in such high densities that a single specimen of the largest species in the genus (B. pitanga) can be found in less than 1 m² (Oliveira 2013). Moreover, Brachycephalus species in the B. didactylus group, including B. hermogenesi and B. dacnis are known to occur at very high densities, potentially even higher than B. pitanga (based on data from myself and other researchers). Additionally, finding Brachycephalus calling under the leaf litter is an extremely challenging task. Therefore, following the concerns raised by Condez et al. (2017), we cannot assume that a calling male is the same specimen collected from under the leaf litter. Consequently, it cannot be conclusively established that the 6.95 mm specimen is an adult male, and, thus, the assertion that B. dacnis is the second smallest vertebrate in the world lacks sufficient evidence. This is a significant claim that should not be sustained in the manuscript based on such fragile evidence of maturity for a 6.95 mm specimen. It is therefore imperative that undissected specimens be considered of indeterminate sex, and that the 6.95 mm specimen be treated as a juvenile until new evidence of maturity in small specimens is obtained through field research. In light of these considerations, I recommend that the authors remove the miniaturization claim from the manuscript.


References
Condez, T.H., Monteiro, J.P.C. & Haddad, C.F.B. (2017) Comments on the current taxonomy of Brachycephalus (Anura: Brachycephalidae). Zootaxa, 4290 (2), 395–400.

Oliveira EG (2013) História natural de Brachycephalus pitanga no núcleo Santa Virgínia, Parque Estadual da Serra do Mar, estado de São Paulo. MSc dissertation, Universidade Estadual Paulista “Júlio de Mesquita Filho”, Rio Claro, 78 pp.

Ribeiro, L.F., Bornschein, M.R., Belmonte-Lopes, R., Firkowski, C.R., Morato, S.A.A. & Pie, M.R. 2015. Seven new microendemic species of Brachycephalus (Anura: Brachycephalidae) from southern Brazil. PeerJ 3: e1011.

5) Bioacoustic analysis.
The authors have now indicated in the manuscript that they employed a note-centered approach to describe the advertisement call of the new species. This method is the same one used by Bornschein et al. (2021) to describe aspects of the call of Brachycephalus sp. Both species are suggested to be the same based on the current manuscript (I was agreeing with this association in my first reply - R1). However, Brachycephalus sp. was described as having more than 37 notes per call, whereas B. dacnis, with the advertisement call revised in R2, has only 1 or 2 notes per call. These significant differences raise doubts about the identity of the species, suggesting that they might not be the same. This implicate a more in-depth bioacoustic analysis and the provision of evidence demonstrating that B. dacnis and Brachycephalus sp., from Bornschein et al. (2021), are indeed the same species. It is advisable that authors carry out their own analyses of the recordings of Brachycephalus sp. and seek additional recordings from other researchers. I am aware of at least one researcher who traveled to Ubatuba specifically to record Brachycephalus sp. after the publication of the 2021 study.
Regarding this subject, I would like to highlight two additional aspects. I had previously suggested that the authors upload the raw call recordings of B. dacnis as part of the review process (see question #1 - R1), which would have enhanced the quality of the review. The authors responded that the recordings are deposited in a public institution and can be accessed upon request. However, this arrangement does not meet the requirements for facilitating access during the review process. Therefore, my review on this matter is compromised, making it almost a blind review.
The second aspect is related to the inclusion of a specific figure showing call parameters (see question #43 - R1). Call parameters are the only source of distinction between the new species and B. hermogenesi, for example. Including this figure will facilitate future research by providing a basis for comparisons between these cryptic species.

Reference
Bornschein, M.R., Ribeiro, L.F., Teixeira, L., Belmonte-Lopes, R., Moraes, L.A. de, Corrêa, L., Maurício, G.N., Nadaline, J. & Pie, M.R. 2021. A review of the diagnosis and geographical distribution of the recently described flea toad Brachycephalus sulfuratus in relation to B. hermogenesi (Anura: Brachycephalidae). PeerJ 9: e10983.

Here, I present other important aspects that I observe in the manuscript.

6) Taxonomic discussion.
To begin, I should clarify that I did not make this comment in R1, which I indicated as partial, because I did not see the need to address it until the new species proposed was properly diagnosed in relation to B. hermogenesi.
The authors raise doubts about the validity of species pairs based on phylogenetic results. Phylogenies like the one presented in the manuscript have been included in many recent descriptions of new Brachycephalus species, and similar results have emerged in those studies. It is not new that phylogenies, sometimes using the same sequences available in GenBank, have indicated similarities or even apparent non-monophyly between species in the B. ephippium and B. pernix groups.
To clarify these taxonomic issues, phylogenomic studies with datasets using 2386 UCE loci (Pie et al. 2018, 2019) were conducted, one of which employed formal species delimitation methods and approaches based on coalescent theory (Pie et al. 2019), and both supported the validity of the species within the B. pernix group. Surprisingly, these studies are not cited in the authors’ discussion.
In summary, the discussion presented by the authors does not introduce new information and, on the contrary, may confuse readers due to the omission of important literature. I have already made several comments in R1 regarding the manuscript’s shortcomings due to a lack of engagement with relevant literature on the group.
Finally, I ask the authors if I have missed any important information, as it is my understanding that B. olivaceus is green and B. mariaeterezae is yellow with a blue dorsal line, contrary to the following claim in the text:

“...the analysis also suggests that some bright colored species need to be revised... Those are the cases of (i) B. olivaceus and B. mariaeterezae, ...”

References
Pie, M.R., Faircloth, B.C., Ribeiro, L.F., Bornschein, M.R. & McCormack, J.E. 2018. Phylogenomics of montane frogs of the Brazilian Atlantic Forest is consistent with isolation in sky islands followed by climatic stability. Biological Journal of the Linnean Society 125: 72-82.

Pie, M.R., Bornschein, M.R., Ribeiro, L.F., Faircloth, B.C. & McCormack, J.E. 2019. Phylogenomic species delimitation in microendemic frogs of the Brazilian Atlantic Forest. Molecular Phylogenetics and Evolution 141: 106627.

7) I strongly suggest that the authors include an analysis of the coloration of specimens of different species in preservative (question #22 - R1, not accepted). “It is expected that many species may not be diagnosable based on coloration in preservative, but it is an opportunity to present this reality to all researchers who will review scientific material in collections. Furthermore, researchers should also be clearly informed if the material that can be identified is limited to live specimens, which implies specific field procedures to document the coloration of live specimens and/or record specimens.” In the authors’ response, they delegated this to a future study, but this analysis is a necessary component in studies describing new species, with minimal or even no interest from journals for a possible future study.

Finally, I have two less critical observations to note:

8) I noticed several writing and formatting errors in the added texts, which require revision (e.g., quotation mark formatting, use of Arabic numerals instead of written numbers, and spelling of words).

9) The abstract does not adhere to the journal’s required format.

Experimental design

Please, see above.

Validity of the findings

Please, see above.

Additional comments

Please, see above.

Reviewer 2 ·

Basic reporting

No comment.

Experimental design

No comment.

Validity of the findings

No comment.

---

## Round 0.3 · accepted · Accept

Thank you for revising your manuscript in line with the review and editor comments. I find that your manuscript is now acceptable for publication in PeerJ.